# Surface-binding molecular multipods strengthen the halide perovskite lattice and boost luminescence

Dong-Hyeok Kim[1,10], Seung-Je Woo [1,10], Claudia Pereyra Huelmo[2,10], Min-Ho Park[1,10], Aaron M. Schankler [2], Zhenbang Dai [2], Jung-Min Heo [1], Sungjin Kim[1], Guy Reuveni[3], Sungsu Kang[4], Joo Sung Kim [1], Hyung Joong Yun[5], Jinwoo Park[1], Jungwon Park [4,6], Omer Yaffe[3], Andrew M. Rappe [2] ✉ & Tae-Woo Lee [1,7,8,9] ✉

Reducing the size of perovskite crystals to confine excitons and passivating surface defects has fueled a significant advance in the luminescence efficiency of perovskite light-emitting diodes (LEDs). However, the persistent gap between the optical limit of electroluminescence efficiency and the photo-luminescence efficiency of colloidal perovskite nanocrystals (PeNCs) suggests that defect passivation alone is not sufficient to achieve highly efficient colloidal PeNC-LEDs. Here, we present a materials approach to controlling the dynamic nature of the perovskite surface. Our experimental and theoretical studies reveal that conjugated molecular multipods (CMMs) adsorb onto the perovskite surface by multipodal hydrogen bonding and van der Waals interactions, strengthening the near-surface perovskite lattice and reducing ionic fluctuations which are related to nonradiative recombination. The CMM treatment strengthens the perovskite lattice and suppresses its dynamic disorder, resulting in a near-unity photoluminescence quantum yield of PeNC films and a high external quantum efficiency (26.1%) of PeNC-LED with pure green emission that matches the Rec.2020 color standard for next-generation vivid displays.

Confining excitons in small perovskite crystals has boosted the radiative recombination of metal halide perovskites (MHPs), enhancing their potential as next-generation light emitters due to their outstanding properties such as narrow emission spectrum, tunable bandgap, and low-cost solution-processability[1–7]. The nanocrystal pinning (NCP) process in polycrystalline bulk perovskites and the synthesis of colloidal perovskite nanocrystals (PeNCs) are methods that effectively confine the exciton within nanocrystals to overcome the intrinsic low exciton binding energy ($E_b$) and long exciton diffusion length of MHPs[8–11]. However, reducing the size of perovskite crystals leads to an increased surface-to-volume ratio, resulting in high susceptibility to surface defects where excitons are quenched by

[1]Department of Materials Science and Engineering, Seoul National University, Seoul, Republic of Korea. [2]Department of Chemistry, University of Pennsylvania, Philadelphia, PA, USA. [3]Department of Chemical and Biological Physics, Weizmann Institute of Science, Rehovot, Israel. [4]School of Chemical and Biological Engineering, Seoul National University, Seoul, Republic of Korea. [5]Research Center for Materials Analysis, Korea Basic Science Institute (KBSI), Daejeon, Republic of Korea. [6]Center for Nanoparticle Research, Institute for Basic Science (IBS), Seoul, Republic of Korea. [7]Institute of Engineering Research, Research Institute of Advanced Materials, Soft Foundry, Seoul National University, Seoul, Republic of Korea. [8]SN Display Co., Ltd., Seoul, Republic of Korea. [9]Interdisciplinary Program in Bioengineering, Seoul National University, Seoul, Republic of Korea. [10]These authors contributed equally: Dong-Hyeok Kim, Seung-Je Woo, Claudia Pereyra Huelmo, Min-Ho Park. ✉e-mail: rappe@sas.upenn.edu; twlees@snu.ac.kr

nonradiative recombination[12–14]. In particular, the highly dynamic nature of ligand binding in colloidal PeNCs causes the photoluminescence quantum efficiency (PLQY) of the spin-coated films to be lower than that of the solutions, further amplifying the vulnerability to surface defects in colloidal PeNCs[15–17]. Accordingly, methods such as varying the surface ligands, doping with cations, and applying surface passivation layers in PeNCs have been all explored to address these drawbacks[18–23]. Furthermore, these strategies have been extensively explored in studies based on cesium lead bromide (CsPbBr$_3$) perovskites, whose emission wavelength ($\lambda_{em}$) is generally lower than 520 nm[15,18,19,21,23]. However, this value is still too far from the $\lambda_{em}$ = 532 nm that is required for the green primary color in the ITU-R Recommendation BT.2020 (Rec. 2020) standard suggested for ultra-high-definition vivid displays[1]. Moreover, the external quantum efficiency (EQE) of colloidal PeNC-light-emitting diodes (LEDs) has not yet reached the theoretical optical limit despite their high PLQY[24,25]. This implies that defect passivation alone up to date is not sufficient for achieving highly efficient colloidal PeNC-LEDs so far. Consequently, further investigation into the underlying mechanisms of luminescence quenching at the surface is necessary to enhance their EQE.

To explore different ways of methods to suppress nonradiative decay, we focused on the dynamic disorder of MHPs and their interaction with excitons and charges in the lattice. MHPs show different characteristics compared to traditional inorganic semiconductors due to their soft lattice structures[26–29]. Conventional Fröhlich-type models and the large-polaron mechanism were not suitable to describe the carrier dynamics of the MHPs, and the dynamic disorder, with a focus on how the strongly anharmonic lattice motions lead to the separation of electrons and holes, has become important in understanding the carrier dynamics of the MHPs[30–33]. Due to the weak ionic bonding in MHPs, organic cations rotate and translate within the inorganic cages and the PbBr$_6$ octahedra make large-amplitude rotations, leading to dynamic disorder of the lattice that affects the trap density, charge transport, and charge recombination. These lattice vibrations can also interfere with the radiative charge carrier recombination process, so

dynamic disorder could cause exciton quenching[34,35]. Furthermore, a recent report using time-resolved mid-infrared spectroscopy suggested that dynamic disorder in MHPs could result in exciton quenching due to electron-lattice interaction[36]. However, despite the importance of dynamic lattice disorder, little research has addressed strategies to mitigate its potential consequences on the luminescence efficiency of perovskite LEDs (PeLEDs).

Here, we demonstrate that π-conjugated molecular multipods (CMMs) with large surface area and multiple separate surface-binding sites (multipods) can suppress the dynamic disorder of the MHP surface (Fig. 1a), leading to near-perfect emitters that minimize nonradiative recombination. We find that CMMs strongly adsorb on the perovskite surface mainly by van der Waals (vdW) interactions and hydrogen bonding and limit lattice motion near the perovskite surface, and thereby reduce the dynamic disorder. As a result, the screening of the charge carriers is effectively reduced, leading to a near-unity PLQY and the high efficiency PeLEDs using colloidal PeNCs without light extraction techniques, with an EQE of 26.1%. In addition, the electroluminescence (EL) emission of our pure green LED using lattice-strengthened PeNCs with a CMM shows a peak at 531 nm and a Commission Internationale de l'éclairage (CIE) 1931 color coordinate of (0.199, 0.762), which approaches the green primary color in the Rec. 2020 standard. These results demonstrate the strong potential of CMM-embedded PeNCs as efficient next-generation pure green emitters to develop ultra-high-definition vivid displays[1].

## Results

### Enhanced luminescence by CMMs

We developed a halide perovskite material system that is composed of halide perovskites surrounded by CMMs, which can control the dynamic disorder at the surface that is closely related to nonradiative recombination. We incorporated CMMs with delocalized π electrons and various electron-transporting functional groups such as 1,3,5-Tris(1-phenyl-1H-benzimidazol-2-yl)benzene (TPBi), 2,4,6-tris[3-(diphenylphosphinyl)phenyl]−1,3,5-triazine (PO-T2T), and tris[2,4,6-

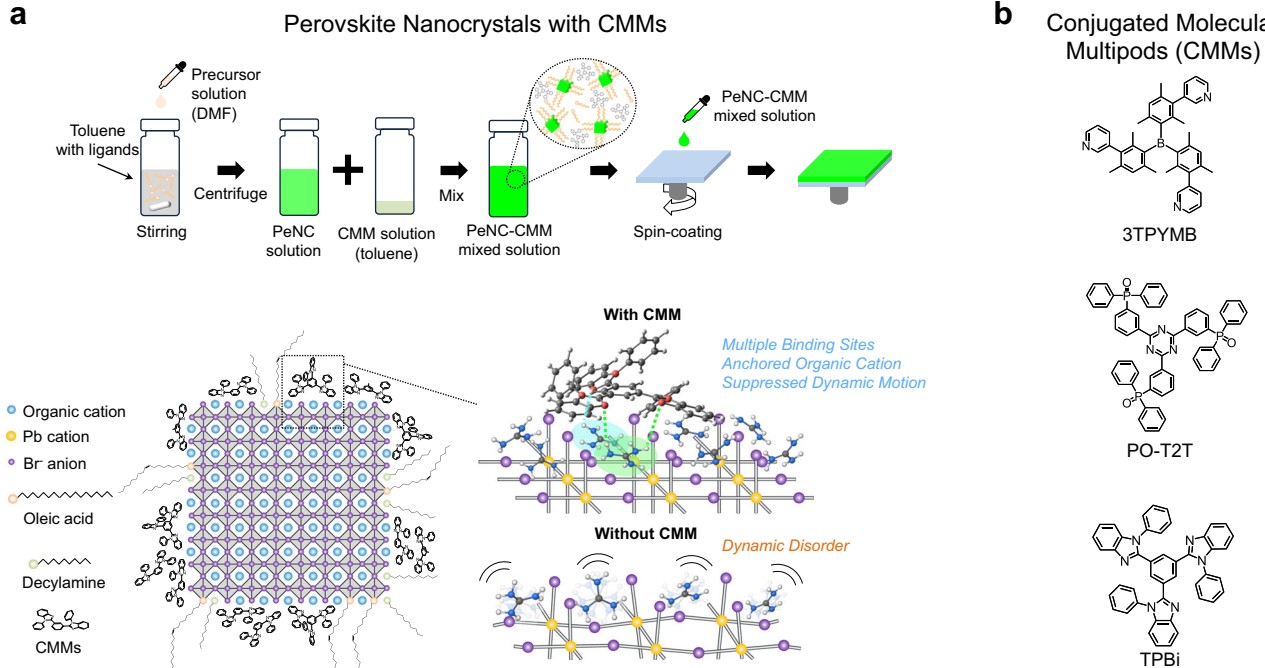

**Fig. 1 | Design of highly efficient light-emitting materials based on PeNCs and surface-binding CMMs. a** Suppressed dynamic motion of PeNC surface by surface-binding CMMs and the fabrication process of PeNC films incorporating CMMs. Multipodal hydrogen bonds between CMM and organic cations are indicated as green and cyan dashed lines. **b** Molecular structure of CMMs.

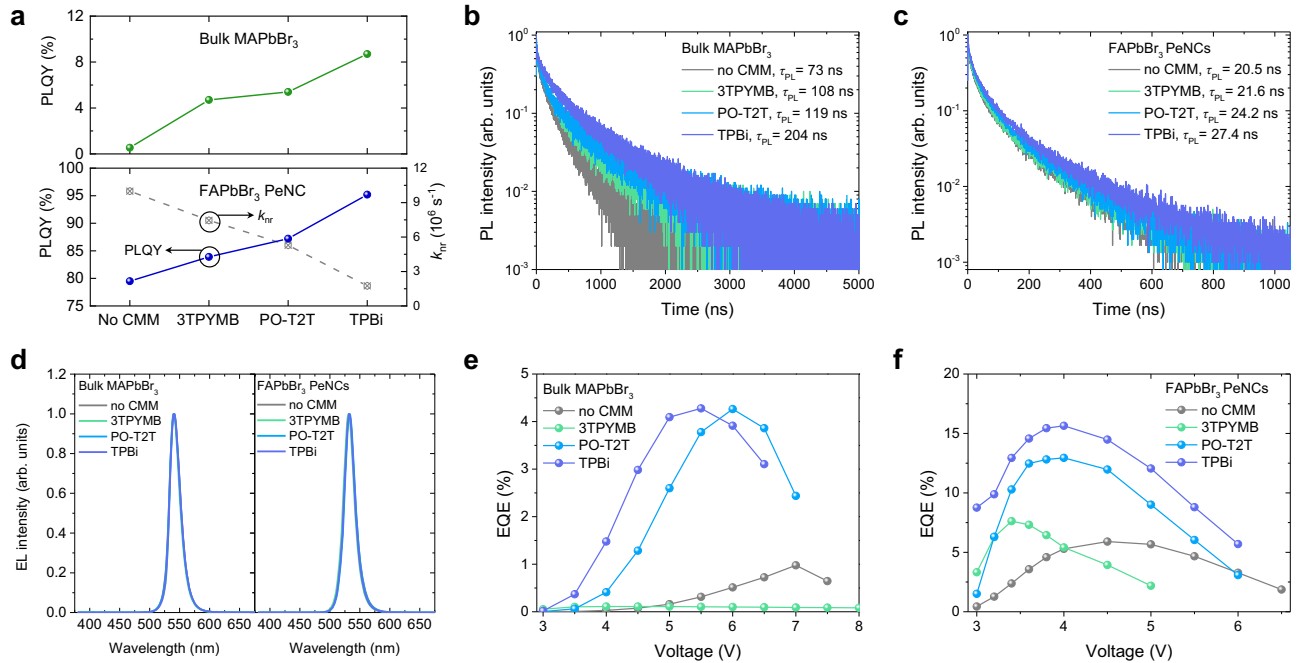

**Fig. 2 | PL and EL characteristics of bulk MAPbBr₃ and FAPbBr₃ PeNCs with CMMs. a** PLQY of bulk polycrystalline MAPbBr₃ and FAPbBr₃ PeNC films. Nonradiative decay rates ($k_{nr}$) are also shown for FAPbBr₃ PeNC films. **b** Transient PL decay of bulk MAPbBr₃ films. **c** Transient PL decay of FAPbBr₃ PeNC films. Legends show average PL decay lifetimes. **d** EL spectra of fabricated devices. **e** EQE curve of devices based on bulk MAPbBr₃. **f** EQE curve of devices based on FAPbBr₃ PeNCs.

trimethyl-3-(pyridin-3-yl)phenyl]borane (3TPYMB) at the surface of the MHPs (Fig. 1b). Hole-transporting molecules were excluded to avoid exciplex formation with the electron-transporting layers and parasitic emission (electromer) from the hole-transporting materials in the EL devices (Supplementary Figs. 1–3). The fabrication process of PeNC films that incorporate CMMs is described in Fig. 1a. Colloidal formamidinium (FA) lead bromide (FAPbBr₃) PeNCs were synthesized under ambient conditions[20]. The colloidal FAPbBr₃ PeNC solution and the CMM solution were mixed with a volume ratio of 10:1 and then spin-coated on substrates. The actual composition of CMMs in the PeNC films was determined to be 10 vol% by analyzing the thickness and refractive index of TPBi, PeNC, and mixed films using variable angle spectroscopic ellipsometry (VASE) (Supplementary Fig. 4). On the other hand, for the fabrication of bulk polycrystalline methylammonium (MA) lead bromide (MAPbBr₃) films incorporated with CMMs, CMM solutions were dropped on the substrate during the crystallization process of the perovskite, so that CMM molecules could locate along the grain boundaries (Supplementary Fig. 5).

Photoluminescence (PL) characteristics of perovskite films with CMMs were analyzed by steady-state PL measurement and time-correlated single-photon counting (TCSPC) measurement (Fig. 2a–c and Supplementary Fig. 6). The PLQYs of both the FAPbBr₃ PeNC and bulk MAPbBr₃ films were improved by incorporating CMMs to encapsulate individual perovskite nanograins and PeNCs, most significantly with the TPBi multipod, followed by PO-T2T (Fig. 2a). The average PL decay lifetime became longer and nonradiative decay rates were decreased, indicating that CMMs suppressed exciton quenching, both for PeNCs and for nanograined bulk perovskite films (Fig. 2a–c, Supplementary Fig. 7, and Supplementary Table. 1). For example, the average PL decay lifetime of FAPbBr₃ PeNC films increased from 20.5 ns without CMM, to 21.6 ns in the film with 3TPYMB, 24.2 ns in the film with PO-T2T, and 27.4 ns in the film with TPBi. Accordingly, the nonradiative decay rates ($k_{nr}$) of FAPbBr₃ PeNC films decreased from $10.0 \times 10^6 \, s^{-1}$ without CMM to $7.45 \times 10^6 \, s^{-1}$ with 3TPYMB, $5.29 \times 10^6 \, s^{-1}$ with PO-T2T, and $1.75 \times 10^6 \, s^{-1}$ with TPBi.

To further study the effect of CMM incorporation on the luminescence properties of perovskites, we fabricated PeLED devices based on CMM-embedded perovskite films as emitting layers (Fig. 2d–f and Supplementary Fig. 8). The devices showed EL spectral line shapes that were independent of the embedded CMMs, indicating confined exciton formation within the perovskite emitting layer for all devices (Fig. 2d). In the case of bulk MAPbBr₃ devices, EQE was enhanced to 4.28% with TPBi and 4.26% with PO-T2T, from 0.98% for the device without CMM. However, the incorporation of 3TPYMB resulted in a decreased EQE (0.11%). The decrease was likely due to aggregation during the NCP process, resulting from the low solubility of alkyl-rich 3TPYMB in the polar dimethyl sulfoxide (DMSO) solvent[37], thus leading to poor charge injection[38] (Fig. 2e and Supplementary Fig. 8a–c). The CMMs also increased the EL efficiency of the colloidal FAPbBr₃ PeNC devices, showing an EQE as high as 15.65% for TPBi, followed by 12.95% for PO-T2T and 7.62% for 3TPYMB, compared to the 5.9% efficiency of the system without CMM (Fig. 2f and Supplementary Fig. 8d–f). The EQE increase of colloidal FAPbBr₃ PeNC devices follows the increasing trend of the PLQY of FAPbBr₃ PeNC films, indicating that the suppression of exciton quenching by the CMMs is also effective in EL.

## Effect of CMMs on perovskite lattice dynamics

To understand the experimental findings of remarkably improved PL and EL performance due to the incorporation of CMMs, we performed a detailed study of the mechanism and consequences of CMM interaction with the perovskite surface. We first modeled systems consisting of a defect-free FAPbBr₃ slab and a single adsorbed CMM using density functional theory (DFT) with the vdW-DF exchange-correlation functional[39]. Each simulation cell contained more than 600 atoms, making these studies a significant computational undertaking, but the large cell is crucial to capturing the multi-site binding of the CMMs. We used structural relaxations and short molecular dynamics trajectories to explore the complex energy landscape of the MHP-CMM bound system. All CMMs showed a clear preference to lie flat on the perovskite surface in order to maximize favorable noncovalent binding

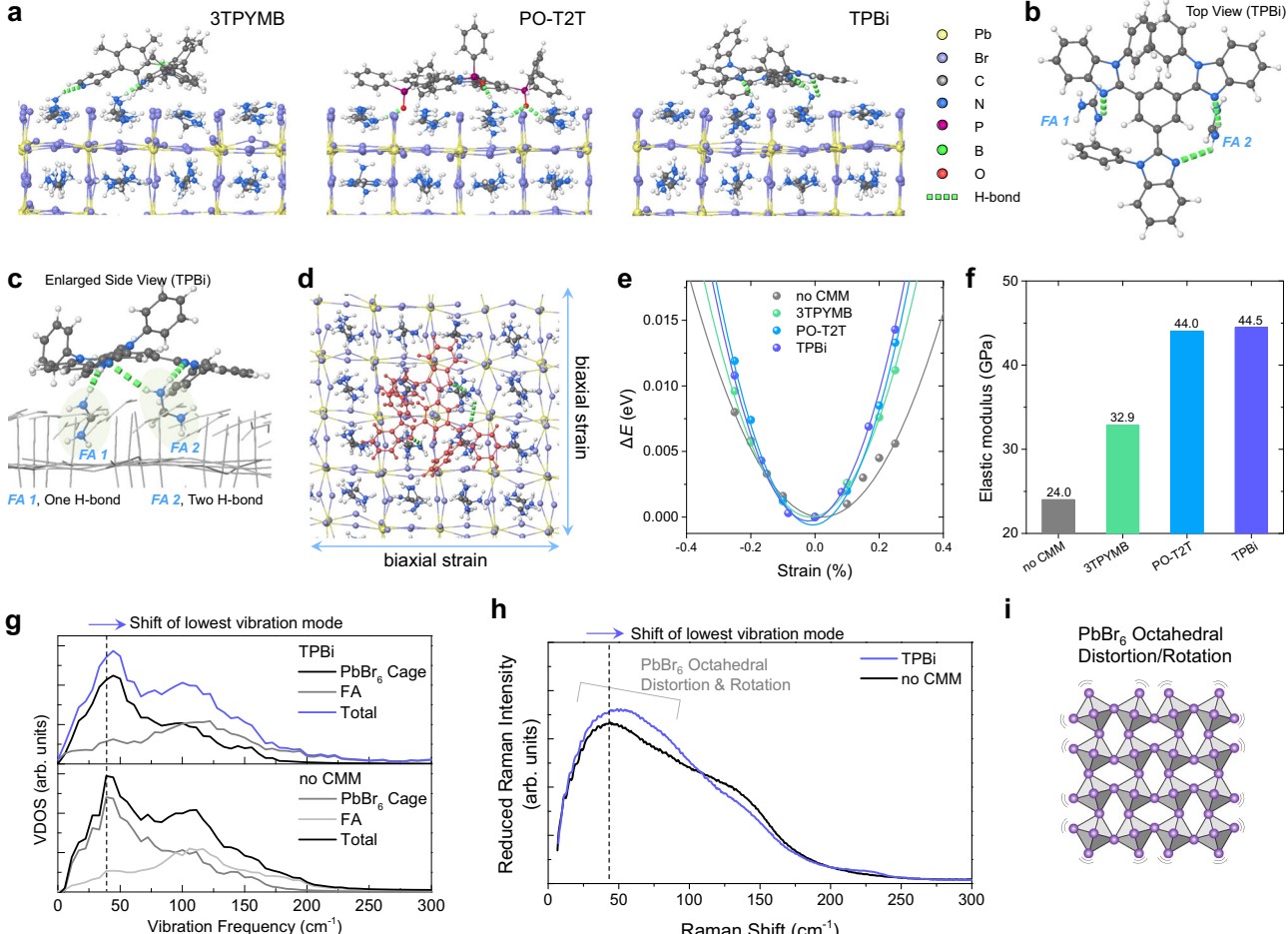

**Fig. 3 | Lattice-strengthened FAPbBr₃ perovskite by CMMs. a** Optimized structures of FAPbBr₃ perovskite with CMMs using DFT calculations. Hydrogen bonds between CMMs and FA cations are shown in green dashed lines. **b** Top view of hydrogen bonds between TPBi CMM and FA cations in the perovskite lattice. **c** Enlarged side view of hydrogen bonds between TPBi CMM and FA cations in the perovskite lattice. Inorganic cages and other FA cations are shown as gray sticks for clarity. FA1 and FA2 form one and two hydrogen bonds with a surface-binding TPBi molecule, respectively. **d** Top view of optimized TPBi-FAPbBr₃ perovskite system and directions of biaxial strain. TPBi molecule is shown in red. **e**, **f** DFT-calculated energy versus strain curves (**e**) and elastic modulus (**f**) values for FAPbBr₃ with CMMs. **g** DFT-simulated low-frequency VDOS for perovskite slab with and without TPBi CMM. **h** Experimentally measured low-frequency Raman spectra of FAPbBr₃ PeNC films with and without TPBi. **i** Distortion and rotation of PbBr₆ octahedral cage.

interactions (Fig. 3a–c and Supplementary Fig. 9), which include both vdW interactions due to the large area covered by the CMMs and hydrogen bonding with surface FA molecules. Due to their multipodal structure, the CMMs bind via their nitrogen and oxygen moieties to more than one place on the perovskite surface. For example, an optimized structure for surface-bound TPBi shows three hydrogen bonds being formed to two different FA cations (Fig. 3b, c). This multi-site binding results in highly favorable adsorption energies (up to 5 eV for TPBi and 7 eV for PO-T2T), so surface-bound CMMs could make a considerable impact on the overall device properties.

With a clear picture of how the CMMs interact with the PeNC surface, we next consider how this binding affects the properties of the perovskite itself. Halide perovskites are known to have soft bonds and a highly dynamic lattice[26–29]. We propose that the multi-site anchoring of the CMMs can help stiffen the surface and suppress lattice motion. To examine this hypothesis, we modeled computationally the effect of the CMM on the elastic properties of thin perovskite slabs. We expect that a net stabilization of the area under the CMM will result in a slab with stiffer in-plane elastic moduli. This is not to suggest that the CMMs make a global change to the properties of the PeNCs, but rather that they suppress dynamics near the surface, where most non-radiative recombination occurs (and which can constitute much of the volume of small PeNCs). To measure the elastic modulus, we applied uniform biaxial strain (Fig. 3d) to each slab geometry to calculate the energy cost of the deformation. The second derivative of the resulting energy-strain curve (Fig. 3e) is proportional to the elastic modulus[40]. The calculations show an increase in the elastic modulus in all CMM-treated surfaces compared with a bare perovskite slab (Fig. 3f), which supports the hypothesis that the soft MHP structure near the surface can be strengthened through interactions with CMMs. The degree of strengthening varies by CMMs, and the trend agrees with observations of luminescent properties, with TPBi and PO-T2T treated surfaces showing the largest enhancement to the elastic modulus.

While a change in the elasticity of the MHP provides a strong indication of lattice strengthening by the adsorbed CMM, we also studied the lattice dynamics more directly to confirm this interpretation. To do so, we collected 15 ps of ab initio molecular dynamics (AIMD) trajectories for each MHP-CMM system (Fig. 3g), and then computed the vibrational density of states (VDOS) from the Fourier transform of the velocity autocorrelation function. Analysis of the low-frequency vibrations shows three peaks, where the first at around 50 cm⁻¹ arises primarily from octahedral rotations and distortions of the inorganic lattice, while the second and third peaks also have substantial contributions from the FA cations[41]. Following the insight

gained from calculations of elastic properties, we expect the adsorbed CMMs to strengthen the lattice near the surface. As stiffened bonds vibrate at a higher frequency, a blue shift in the VDOS is indicative of this lattice strengthening. In fact, we indeed observed a slight blue shift in the $50 \, cm^{-1}$ peak of the CMM-treated system relative to the untreated MHP (Fig. 3g and Supplementary Fig. 10), which agrees with the elastic modulus results.

To further confirm this behavior, we performed low-frequency Raman measurements of TPBi-treated PeNCs (Fig. 3h) to experimentally investigate how vibrational properties are modified by the presence of CMMs. Like the computationally predicted VDOS, the Raman spectrum shows three features at low frequency. As the VDOS is not weighted by the Raman cross-section, the qualitative agreement in peak positions indicates that our computational model system provides a reliable picture of the PeNC lattice dynamics. The lowest frequency peak measured in Raman showed a slight shift to higher frequency, again supporting the theoretical predictions that CMMs suppress the large-amplitude, low-frequency dynamics in FAPbBr₃. As has been demonstrated previously, this dynamic disorder of the perovskite lattice screen injected charges, reducing the carrier mobility and impeding the probability of radiative recombination[30–32]. As a stiffer lattice in general have a lower amplitude of dynamical motion, charge trapping/screening will be less effective, and so nonradiative recombination will be reduced, thereby enhancing radiative emission. We suggest that the observed increase in PLQY and EQE of bulk MAPbBr₃ and FAPbBr₃ PeNC films and devices can be attributed to the anchoring of the CMMs on the MHP surface, which we have revealed through first-principles calculation and Raman spectroscopy.

## Interaction between perovskites and CMMs

To evaluate alternative explanations for the improved luminescent properties, we studied whether CMM incorporation has additional effects beyond strengthening the near-surface lattice. To determine whether CMMs influence the crystal structure of perovskites, we conducted X-ray diffraction (XRD) measurement (Fig. 4a and Supplementary Fig. 11). The FAPbBr₃ PeNCs with CMMs showed the same XRD patterns that correspond to the cubic phase[20], indicating that the incorporation of CMMs does not induce any change in the crystal structures. Transmission electron microscopy (TEM) images of colloidal FAPbBr₃ PeNCs showed average particle size of ~9.78 ± 0.23 nm without CMM, 9.90 ± 0.21 nm with 3TPYMB, 9.98 ± 0.20 nm with PO-T2T, and 9.88 ± 0.24 nm with TPBi (Fig. 4b and Supplementary Fig. 12). This result indicates that post-synthesis blending of CMMs into colloidal PeNCs has negligible effect on the average particle size and distribution. Furthermore, CMM incorporations caused no significant change in the morphology of perovskite films (Supplementary Figs. 13 and 14). Combined analysis of ultraviolet photoelectron spectroscopy (UPS) spectra and UV-vis absorption spectra confirmed that all CMMs do not affect the valence band maximum (VBM) or conduction band maximum (CBM) of PeNCs (Fig. 4c and Supplementary Figs. 15 and 16). The work functions were not changed by TPBi and 3TPYMB CMMs, but the incorporation of PO-T2T CMM lowered the work function by 0.17 eV, making FAPbBr₃ PeNC with PO-T2T slightly more n-type than the FAPbBr₃ PeNC without CMM. However, the effect of CMMs on the current density of the electron-only device was negligible despite the electron-transporting nature of the CMMs (Supplementary Fig. 17). On the other hand, due to the hole-blocking characteristics of the CMMs, the current density of the hole-only device was largely decreased (Supplementary Fig. 17). This led to the decreased current density of colloidal FAPbBr₃ PeNC-LEDs with PO-T2T and TPBi CMMs compared to that of the device without CMM. Thus, as the CMMs do not significantly impact the crystal structure, particle size distribution, film morphology, VBM, and CBM of PeNCs, the lattice strengthening effect is the primary contributor to the improved luminescent efficiency.

To experimentally confirm the mode of interaction between the CMMs and the perovskite, solution nuclear magnetic resonance (NMR) analysis was conducted using FABr, dissolved with PO-T2T or TPBi in DMSO-d6 (Fig. 4d). The −NH₂ signals in FABr were obviously shifted downfield when TPBi or PO-T2T were included. This result indicates the formation of hydrogen bonds between FA cation and the nucleophilic functional groups that have lone pair electrons (benzimidazole and phosphine oxide group) in the CMMs[42,43]. To confirm this effect between the CMMs and the FA cations at the surface of the perovskite crystal lattice, we measured attenuated total reflectance-Fourier transform infrared (ATR-FTIR) spectra of FAPbBr₃ and FAPbBr₃-TPBi (Fig. 4e). Addition of TPBi caused decrease in wavenumber and peak broadening of both the symmetric N-H stretching mode ($v_s$(N−H)) and the C=N stretching mode ($v$(C=N)); these changes clearly shows the hydrogen bonding formation between FA cation and TPBi[43,44]. X-ray photoelectron spectroscopy (XPS) analysis of Br 3$d$ states in CMM-mixed FAPbBr₃ films shows no evidence of strong interaction between CMMs and the surface Br ions (Supplementary Fig. 18). Together, these results confirm our theoretical predictions and demonstrate that CMMs are adsorbed to the surface of the PeNCs by hydrogen bonding with FA cations.

## Effect of CMMs on GA-doped FAPbBr₃ PeNCs

To further increase the device efficiency, we consider covering guanidinium (GA)-doped FAPbBr₃ PeNCs with CMMs. GA doping in PeNCs leads to partially GA-terminated surfaces, which is expected to enhance hydrogen bonding with CMMs and increase the surface area covered by CMMs[20]. Solution ¹H-NMR analysis showed a downfield shift of the -NH₂ signals in GA cations blended with TPBi or PO-T2T, indicating that GA cations also form hydrogen bonding with CMMs (Fig. 5a). Similar to the studies of FAPbBr₃ PeNCs (Fig. 4a, b and Supplementary Figs. 12 and 13), CMM incorporation had negligible impact on the crystal structure, film morphology, and particle size of GA-doped FAPbBr₃ PeNCs (Supplementary Figs. 19–21). We performed DFT calculations to analyze the interaction between GA-terminated FAPbBr₃ perovskite surface and the TPBi molecule, which is the CMM treatment that yielded the highest efficiency performance in MA and FA-based devices (Fig. 5b–d). In line with the results observed for the FAPbBr₃ perovskite, TPBi adsorbed horizontally to the GA-terminated surface of the perovskite (Fig. 5b). Moreover, we confirmed that TPBi has stronger interaction (+0.3 eV) with the GA-terminated perovskite surface than the FAPbBr₃ perovskite surface because GA has an additional amine group that can make a hydrogen-bond[20]. In addition, the energy-strain curves of GA-terminated perovskite also show a slightly increased elastic modulus when TPBi is added (Fig. 5b–d), indicating that TPBi serves to strengthen the surface lattice. Using TPBi as CMM on the FA₀.₉GA₀.₁PbBr₃ PeNC film significantly increased the PLQY to near unity, and the PL decay lifetime also increased (Fig. 5e and Supplementary Figs. 22–25). Next, we fabricated LED devices of colloidal FA₀.₉GA₀.₁PbBr₃ PeNCs incorporating CMMs using the same method as for colloidal FAPbBr₃ PeNC devices (Fig. 5f–j and Supplementary Figs. 26, 27). The CMM treatment on GA-doped PeNCs results in a more efficient colloidal PeNC-LED with an EQE of 26.1%, calculated based on angular EL distribution[45] (Supplementary Fig. 28). Our colloidal PeNC-LEDs based on this materials design approach achieved the high efficiency in PeLEDs using colloidal PeNCs without an outcoupling enhancement technique (Fig. 5k, Supplementary Fig. 29, and Supplementary Tables. 2, 3). For practical use, pure green (CIEy > 0.75) emission with high efficiency is of critical importance. Our device exhibited pure green EL emission (full width at half maximum ~ 20 nm) at 531 nm and the CIE 1931 color coordinate of (0.199, 0.762), closely approaching the green primary color (532 nm) in the Rec. 2020 standard[1] (Fig. 5k and Supplementary Table. 2). The narrow distribution of the EQE histogram of TPBi CMM devices shows that our lattice-strengthening strategy is effective in enhancing the EL efficiency of the

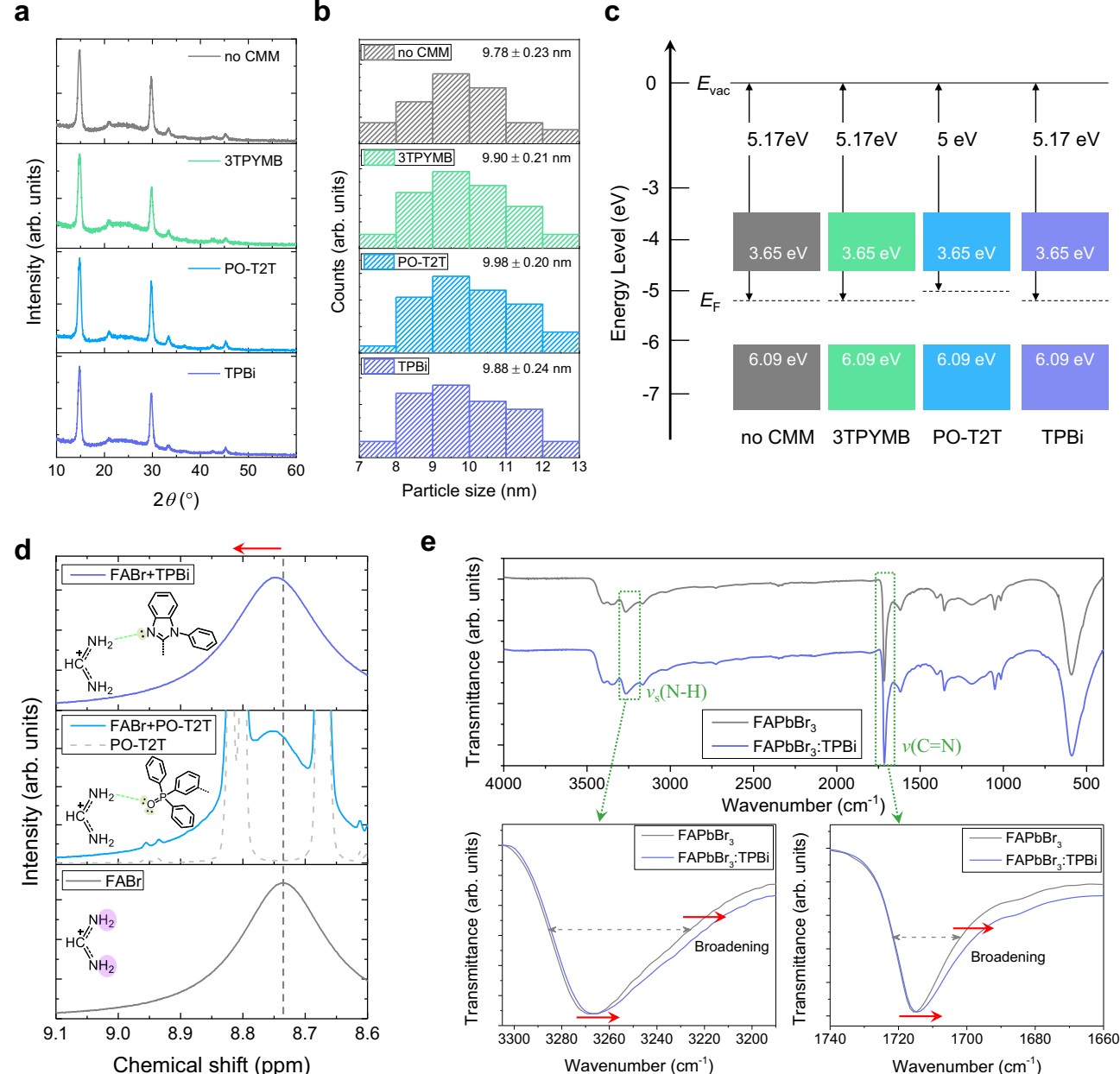

**Fig. 4 | Interaction between FAPbBr₃ PeNC and CMMs. a–c** Independent characteristics of FAPbBr₃ PeNCs with CMMs. Crystal structure (**a**) of FAPbBr₃ PeNCs. XRD patterns of FAPbBr₃ PeNC films. The incorporation of CMMs did not cause a peak shift. The size distribution (**b**) of colloidal FAPbBr₃ PeNCs. Particle size was extracted from TEM images. Average particle size and standard error were calculated. Electronic structure (**c**) of FAPbBr₃ PeNCs. Energy band diagrams of FAPbBr₃ PeNCs with CMMs were obtained by combining UPS analysis and optical bandgap. VBM and optical bandgap were derived from UPS spectra and UV-vis absorption spectra, respectively. The CBM was calculated based on the VBM level and optical bandgap of PeNCs. **d, e** Hydrogen bonding formation between CMMs and perovskite. The ¹H·NMR spectra (**d**) of FABr and mixtures of FABr, TPBi and PO-T2T. −NH₂ signal of FABr marked with violet color on FABr molecule. The ¹H·NMR spectrum of pure PO-T2T is shown in the dashed line. Excluding the peaks arising from PO-T2T, the peak shift of the −NH₂ signal in FA can be identified. Hydrogen bonding between FA cation and CMMs is indicated by a green dashed line. ATR-FTIR spectra (**e**) of FAPbBr₃ and FAPbBr₃-TPBi without ligands. The regions of symmetric N-H stretching mode ($v_s$(N−H)) and C=N stretching mode ($v$(C=N)) are magnified in the graph below.

PeLEDs with high reproducibility (Fig. 5i). Moreover, the fabricated device showed a high EQE (26.1%) close to the theoretical efficiency (29.4%) obtained from optical simulation assuming a perfect charge balance and a PLQY of unity (Fig. 5j and Supplementary Fig. 30). In addition to the improvement of efficiency, the TPBi CMM device showed a longer device half-lifetime by a factor of 3.1 compared to the device without CMM (Supplementary Fig. 31). Furthermore, TPBi CMM also improved photostability of FA₀.₉GA₀.₁PbBr₃ PeNC film (Supplementary Fig. 32). Consistent results showing improvement in EL and PL

stability by a TPBi CMM indicate that lattice-strengthening by TPBi that does not induce ligand detachment also contributes to suppressing the degradation process of the perovskite.

In colloidal PeNCs, the highly dynamic nature of ligand binding also influences the PLQY of colloidal PeNCs[15–17]. Interestingly, we find that unlike TPBi, the incorporation of PO-T2T into the GA-doped perovskite system introduces unwanted side effects that deteriorate the PL and EL efficiencies and cancel out the lattice strengthening effect (Fig. 6). The $E_b$ of FA₀.₉GA₀.₁PbBr₃ PeNCs films determined from

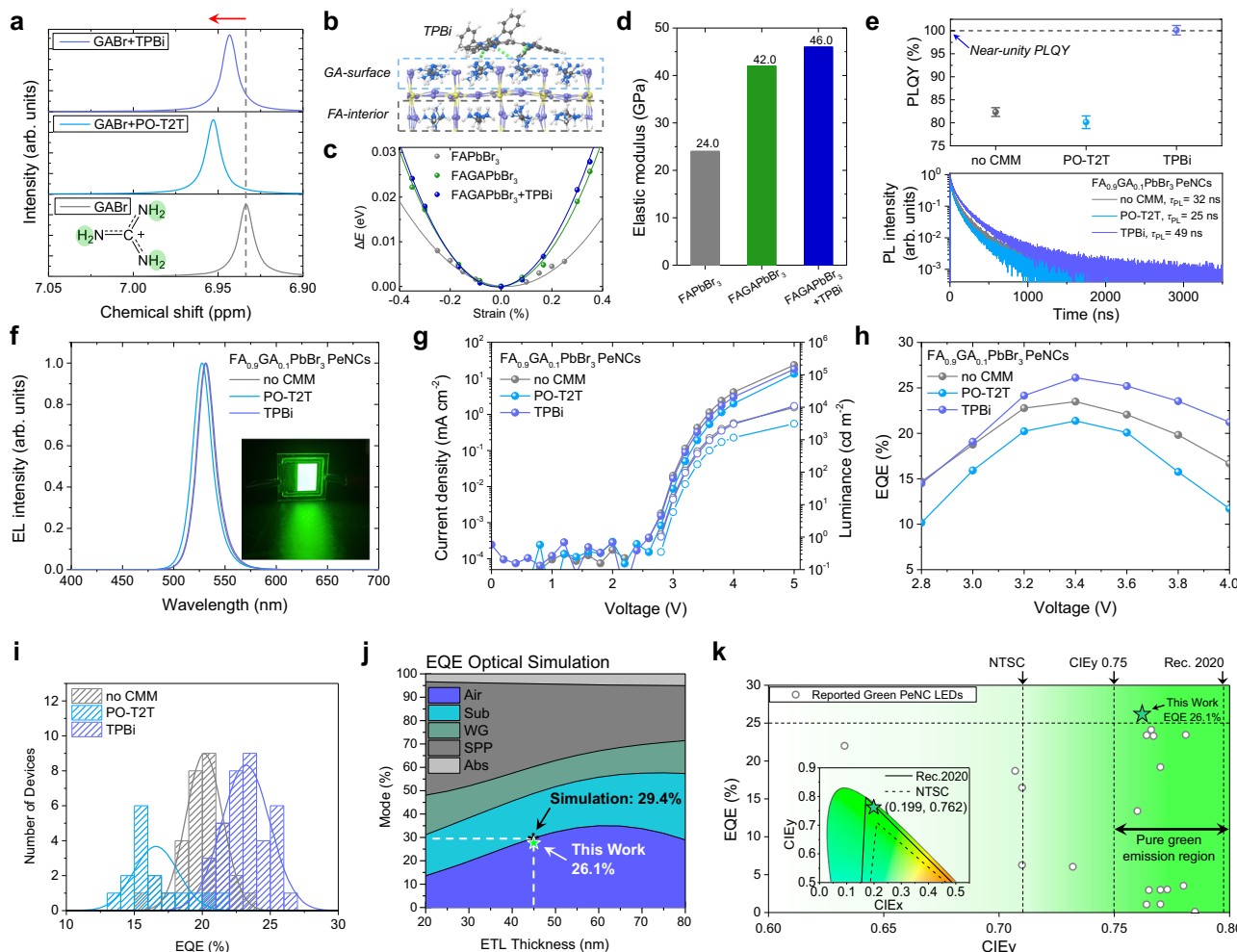

**Fig. 5 | Effect of CMMs on FA$_{0.9}$GA$_{0.1}$PbBr$_3$ PeNCs and their luminescent characteristics. a** $^1$H-NMR spectra of GABr and mixtures of GABr, TPBi, and PO-T2T. −NH$_2$ signal of GABr was marked with green color on the GABr molecule. **b** DFT-optimized structure of TPBi-FAGAPbBr$_3$ perovskite. **c** DFT-calculated energy versus strain curves for FAGAPbBr$_3$ with CMMs. **d** DFT-calculated elastic modulus for FAPbBr$_3$, FAGAPbBr$_3$, and FAGAPbBr$_3$ with TPBi. **e** PLQY and transient PL decay of FA$_{0.9}$GA$_{0.1}$PbBr$_3$ PeNCs. The error bars are the standard error. **f–i** EL characteristics of FA$_{0.9}$GA$_{0.1}$PbBr$_3$ PeNC devices. EL spectrum (**f**), J−V−L curve (**g**), EQE-voltage

curve (**h**), and EQE histogram (**i**). **j** Optical simulation of EQE as a function of electron transporting layer (ETL) thickness. ETL thickness of fabricated devices is 45 nm. The calculated EQE from the optical simulation at an emissive layer (EML) thickness of 30 nm is 29.4%. Air outcoupled mode, Sub substrate guide mode, WG waveguide mode, SPP surface plasmon polariton mode, Abs absorption. **k** Summary of reported EQE according to CIEy of green-emitting colloidal PeNC-LEDs without outcoupling enhancement technique (inset: corresponding color coordinate of our device in CIE 1931 color space).

the Arrhenius fitting of temperature-dependent PL intensity also shows the decreased value of $E_b = 79.2$ meV with PO-T2T CMM compared to $E_b = 101.5$ meV without CMM, whereas FA$_{0.9}$GA$_{0.1}$PbBr$_3$ PeNC with TPBi showed increased $E_b = 181.0$ meV, clearly indicating a strengthened lattice (Fig. 6a, b). Transient EL measurement of the PO-T2T CMM device shows slower rising time after the voltage pulse on and longer delayed EL decay after the voltage pulse off compared to the TPBi CMM device, indicating that defects are generated with the addition of PO-T2T (Supplementary Fig. 33). To further investigate the origin of the detrimental effect of PO-T2T on FA$_{0.9}$GA$_{0.1}$PbBr$_3$ PeNCs, the Pb 4$f$ XPS spectra of bulk MAPbBr$_3$, FAPbBr$_3$ PeNCs, and FA$_{0.9}$GA$_{0.1}$PbBr$_3$ PeNCs with and without CMM was measured (Fig. 6c and Supplementary Fig. 34). For the case of bulk MAPbBr$_3$ and FAPbBr$_3$ PeNCs, there was no formation of metallic Pb when treated with any CMM. This is also the case for FA$_{0.9}$GA$_{0.1}$PbBr$_3$ PeNCs treated with TPBi. On the other hand, when PO-T2T instead of TPBi was incorporated in FA$_{0.9}$GA$_{0.1}$PbBr$_3$ PeNC films, the formation of metallic Pb was clearly observed. This is clear evidence that the surface treatment of GA-doped FAPbBr$_3$ PeNCs with TPBi reduces the

dynamic disorder of the perovskite lattice without leading to luminescence degradation of the colloidal PeNC system. We hypothesize that the performance degradation in the presence of PO-T2T is due to the presence of nucleophilic functional groups with lone pairs that compete with the perovskite or the hydrogen bond to ligands (Fig. 6d) that are also located on the perovskite surface. As we have previously reported, partial substitution of GA cation at the nanocrystal surface makes the binding energy of the bound ligands slightly weaker[20], which facilitates this competition in the GA-doped system. To investigate the interaction between ligands and CMMs, we performed a solution $^1$H-NMR analysis (Fig. 6d–f and Supplementary Fig. 35). When CMMs were mixed with ligand solutions, there was no peak shift in 1(α-CH$_2$) of decylamine (DAm), but the peak for 2(α-CH$_2$) of oleic acid (OA) showed a downfield shift, indicating that only OA interacted with CMMs[46]. From these results, we found that hydrogen of the carboxyl group of OA ligand and lone pairs of CMMs form hydrogen bonds. OA ligand mixed with PO-T2T showed a strong downfield shift in the peaks for α-CH$_2$ of OA. On the other hand, OA mixed with TPBi only showed negligible downfield shift (Fig. 6e). This

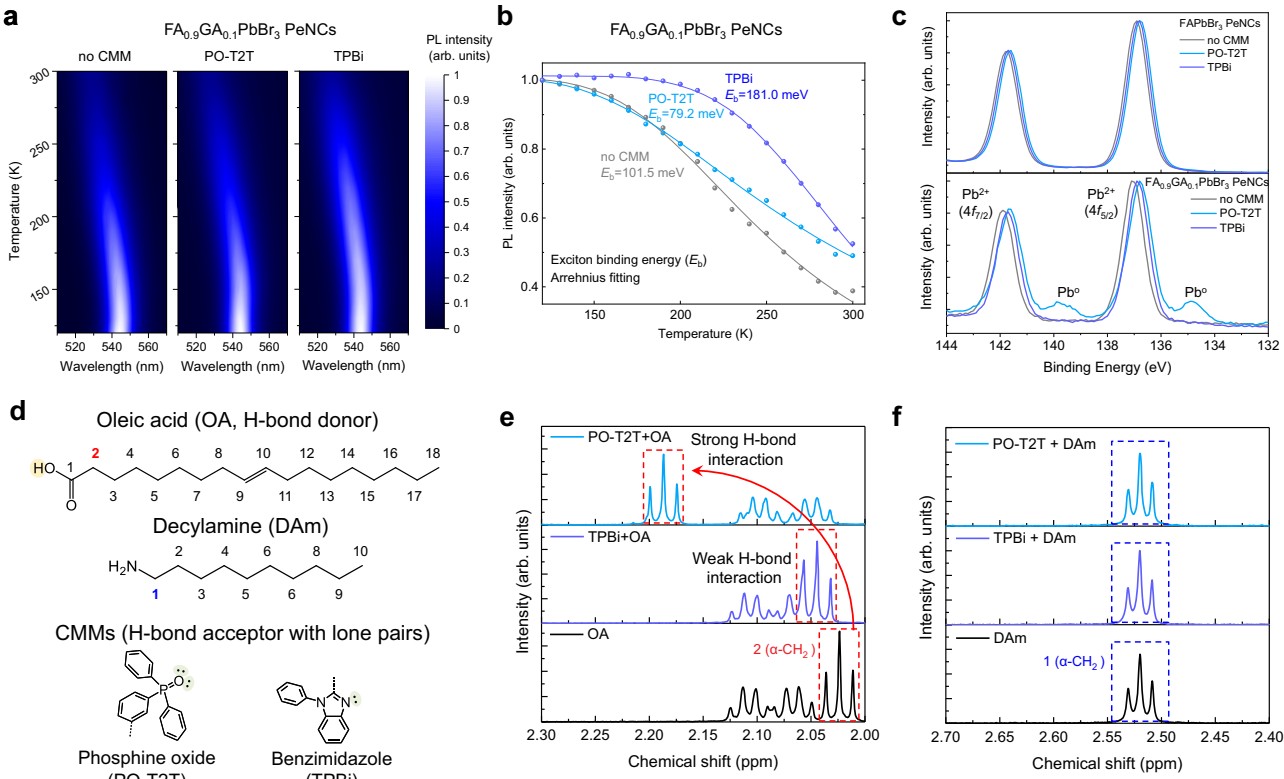

**Fig. 6 | Interaction between CMMs and OA ligand. a, b** Temperature-dependent PL spectra (**a**) and integrated temperature-dependent PL intensity (**b**) of $FA_{0.9}GA_{0.1}PbBr_3$ PeNC films with and without CMMs. The color bar shows the normalized intensity of each film. Experimental data are fitted using the Arrhenius equation to obtain exciton binding energies. **c** Pb 4*f* XPS spectra of $FAPbBr_3$ and

$FA_{0.9}GA_{0.1}PbBr_3$ PeNC films with and without CMMs. **d** Molecular structure of OA, DAm, and hydrogen bond (H-bond) acceptor moieties of CMMs. **e, f** $^1$H-NMR spectra of the mixed solution of CMMs with OA ligand (**e**) and DAm ligand (**f**) in the benzene-d6 solvent.

implies that PO-T2T forms a stronger hydrogen bond with OA than TPBi. We propose that PO-T2T CMM intervenes in the dynamic ligand binding process and forms a hydrogen bond pair with the OA ligand, resulting in the metallic Pb formation at the perovskite (Fig. 6c)[17,46]. The strong interaction between PO-T2T and OA observed from the $^1$H-NMR analysis, the metallic Pb formation, and the reduction in $E_b$ of $FA_{0.9}GA_{0.1}PbBr_3$ PeNC upon incorporation of PO-T2T indicate that this CMM detaches the surface ligand from the PeNCs, generating surface defects (Fig. 6a–c). Consequently, the lattice-strengthening effect and ligand-detachment effect mutually cancel out in $FA_{0.9}GA_{0.1}PbBr_3$ PeNCs with PO-T2T treatment, diminishing PLQY and EQE (Fig. 5d–j). On the other hand, $FA_{0.9}GA_{0.1}PbBr_3$ PeNC films with TPBi did not show metallic Pb formation, which is related to the weaker hydrogen bonding with OA ligand as observed from the $^1$H-NMR analysis. Enabled by the comprehensive understanding of the dynamic disorder of MHP lattice and the dynamic binding behavior of the ligands in colloidal PeNCs, we provide a valuable strategy to further improve the efficiency and stability of PeLEDs based on colloidal PeNCs.

In conclusion, we develop PeLEDs showing enhanced luminescence efficiency by incorporating surface-binding molecular multipods into bulk polycrystalline perovskite and colloidal PeNCs. DFT and AIMD simulations reveal that CMMs are strongly adsorbed on the perovskite surface by multipodal hydrogen bonding and vdW interactions, strengthening the near-surface lattice of MHPs and suppressing dynamic disorder. These findings are further confirmed by the blue-shifted low-frequency Raman spectra of perovskite-CMM film and increased luminescence efficiencies in the films and the devices, implying suppressed nonradiative recombination. We suggest that a lattice-strengthening CMM which effectively suppresses the dynamic

disorder of the MHP lattice while working synergistically with defect passivation strategies without ligand detachment constitutes an effective approach for more ideal perovskite light-emitting materials. Our PeNC with a lattice-strengthening CMM provides a near-unity PLQY, pure green emission characteristics to satisfy the Rec. 2020 standard, and a high EQE of 26.1% in colloidal PeNC-LEDs without light extraction technique. Our pure green colloidal PeNC-LED simultaneously achieves EQE over 25% and CIEy over 0.75. In addition, a lattice-strengthening CMM without ligand detachment also leads to the improvement in the operational lifetime of colloidal PeNC-LEDs by a factor of 3.1 compared to that of no CMM device. Our results demonstrate how CMM adsorption influences and mitigates adjacent MHP lattice dynamics and leads to enhanced PeLED efficiency and stability. More generally, this work adds the important tool of lattice-strengthening strategy with multipodal CMMs to the design space of molecular additives for high-efficiency pure green PeLEDs to realize vivid color displays.

## Method

### Synthesis for colloidal PeNCs

Both colloidal $FAPbBr_3$ and $FA_{0.9}GA_{0.1}PbBr_3$ PeNCs were synthesized under ambient conditions[20]. For colloidal $FAPbBr_3$ PeNCs, the precursor solution was prepared by dissolving FABr (0.2 mmol) and $PbBr_2$ (0.1 mmol) in 0.5 ml of *N,N*-dimethylformamide (DMF). For colloidal $FA_{0.9}GA_{0.1}PbBr_3$ PeNCs, FABr (0.18 mmol) and GABr (0.02 mmol) were dissolved in the same amount of $PbBr_2$ and DMF. The mixture of the non-polar solvent and ligands for reprecipitation was composed of toluene (5 ml), 1-butanol (2 ml), OA (300 µl), and DAm (24.2 µl). The precursor solution (0.15 ml) was dropped into a non-polar solvent mixture under vigorous stirring. After 10 min,

colloidal PeNCs were washed by sequential centrifugation and dispersed in toluene.

## Fabrication of bulk perovskite films

Cleaned silicon or quartz substrates were exposed to UV-ozone treatment for 15 min to make the substrate hydrophilic and remove organic residues. Gradient hole injection layer (GraHIL) films were fabricated by spin coating a solution consisting of poly(3,4-ethylenedioxythiophene):poly(styrene sulfonate) (PEDOT:PSS) and perfluorinated ionomer (PFI). For the fabrication of bulk perovskite films, a precursor solution of $MAPbBr_3$ dissolved in DMSO (1.2 M, $MABr:PbBr_2 = 1.06:1$) was spin-coated on GraHIL film. During the spin coating process at 3000 rpm, CMM solutions that dissolved CMMs in chloroform (CF) solvent were dropped onto the substrate[6]. For the case of no CMM, pure CF solvent was used for the NCP process.

## Fabrication of perovskite nanocrystal films

PeNC-dispersed toluene solution was mixed with the CMM-dissolved toluene solution (5 mM) with a volume ratio of (10:1). Then, the mixed solution was spin-coated on the substrates at 500 rpm for 60 s to form CMM-incorporated PeNC films.

## Fabrication of perovskite EL devices

Fluorine-doped tin oxide (FTO) or indium tin oxide (ITO) (70 nm) were cleaned sequentially by sonication in acetone and isopropanol for 15 min each (Supplementary Fig. 3). Cleaned substrates were exposed to UV-ozone treatment for 15 min to make the substrate hydrophilic and remove organic residues. GraHIL films were fabricated by spin coating a GraHIL solution. Then, bulk perovskite and PeNC EMLs were spin-coated as the same method written above. TPBi (45 nm), LiF (1 nm), and Al (100 nm) were sequentially deposited onto perovskite films in a high-vacuum chamber ($<10^{-7}$ Torr). The devices were encapsulated to prevent exposure to moisture and $O_2$ during measurements.

## EL characterization

The current density-voltage-luminescence ($J-V-L$) characteristics of PeLEDs were measured using a spectroradiometer (CS-2000, Minolta) and an electrical source-measurement unit (Keithley 236). The EQE of PeLEDs was calculated based on the angle-dependent EL profile[45]. Transient EL analysis was performed using a streak camera system composed of a streak scope (C10627, Hamamatsu Photonics), a CCD camera (C9300, Hamamatsu Photonics), a delay generator (DG645, Stanford Research Systems), and a function generator (33220A, Agilent).

## Analysis of PL characteristics

PLQY of the PeNC films were measured with an integrating sphere and PMT detector using 325 nm CW He:Cd laser as the excitation source. The integrating sphere was purged with nitrogen gas during the measurement of PLQY. The PL spectra of encapsulated PeNC films were measured with a CCD spectrometer (Maya2000, Ocean Optics) using 325 nm CW He:Cd laser as the excitation source. PL spectra and PLQY of bulk perovskite films were measured with a JASCO FP8500 spectrofluorometer with a xenon arc lamp with 405 nm as an excitation source. Transient PL decay of perovskite films was obtained by a TCSPC set-up (FluoTime 300 (PicoQuant)) composed of a 405 nm pulsed laser as an excitation source (LDH-P-C-405B, PicoQuant), a photon-counting detector (PMA Hybrid 07) and a TCSPC module (PicoHarp, PicoQuant). The average PL decay lifetimes of bulk perovskite and PeNC films were obtained by fitting transient PL curves using a tri-exponential decay function[47]. The radiative decay rate ($k_r$) and nonradiative decay rate ($k_{nr}$) were calculated based on the relation

below[48] (Eq. 1, 2).

$$PLQY = k_r / (k_r + k_{nr}) \tag{1}$$

$$k_r + k_{nr} = 1/\tau_{PL} \tag{2}$$

$E_b$ was calculated using the Arrhenius equation, $I_T = I_0/(1 + A \exp(-E_b/kT))$, where $I_T$ is the integrated PL intensity, $I_0$ is PL intensity at 0 K, $A$ is constant, $k$ is Boltzmann constant, and $T$ is temperature.

## DFT calculations

DFT calculations using the vdW-DF functional[39] were performed using the Spanish Initiative for Electronic Simulations with Thousands of Atoms (SIESTA) code[49]. This scheme employs a localized atomic basis set in conjunction with the pseudo-potential approximation. Norm-conserving pseudopotentials were selected, the double-zeta basis set with polarization functions was employed, and the mesh cutoff was fixed to 200 Ry. Lattice parameters of $4 \times 4 \times 2$ supercells of cubic and Br-terminated $FAPbBr_3$ were fully optimized in all calculations except when calculating the strain curves. The $FAGAPbBr_3$ perovskite slab was built replacing only the topmost FA cations in the surface. Geometry optimizations were performed until all residual forces were smaller than 0.0008 eV/Å, and the Brillouin zone was sampled at the gamma point, due to the large number of atoms (>600) in these systems. A large vacuum space of 20 Å is included to avoid the interaction between the periodic layers along the $z$-axis. To study the mechanical properties of these systems, we estimated their elastic modulus by taking the second derivative of the energy vs % strain curve. The isolated CMM optimizations were performed at the B3LYP/6-31 G* level of theory with the Q-Chem code[50].

## AIMD simulations

AIMD simulations were performed based on a Nosé-Hoover thermostat canonical scheme (NVT) as implemented in the SIESTA code. The temperature was set to 298 K and the time step set to 1 fs, chosen in accordance with the highest frequency motions of the molecular modes. The instantaneous velocities are determined by the finite difference of the atomic positions, and the VDOS was obtained by taking the Fourier transform of the mass-weighted velocity autocorrelation function, which was computed using the Wiener–Khinchin theorem (Eq. 3).

$$VDOS(\omega) = \mathcal{F}\left[\frac{1}{kT}\sum_i m_i \langle v_i(t+\tau) \cdot v_i(\tau)\rangle_\tau\right] = \frac{1}{kT}\sum_i m_i |\mathcal{F}[v_i(t)]|^2 \tag{3}$$

Here, $m_i$ and $v_i(t)$ are the mass and velocity trajectory of each atom, $k$ is the Boltzmann constant and $T$ is the temperature. Fourier transformation is denoted by $\mathcal{F}[\cdot]$, and $\langle \cdot \rangle_\tau$ indicates averaging over different starting times $\tau$ within the trajectory. Only atoms in the perovskite slab were included in the sum over $i$. For species-resolved VDOS, the sum was further restricted by atomic species.

## Raman spectroscopy

Raman scattering measurements were conducted in a home-built back-scattering system[51,52] using a below band-gap 1.58 eV CW pump-diode laser (Toptica Inc., USA). The incident beam was linearly polarized by a Glan-laser polarizer (Thorlabs, USA), directed into a microscope (Zeiss, USA), and focused on the sample through a 0.55 NA/50× objective (Zeiss, USA). The excitation polarization was controlled by a zero-order half-wave plate (Thorlabs, USA). The back-scattered beam was collected by the objective and passed through another polarizer to collect only light that was scattered either parallel or perpendicular to the incident polarization. Rayleigh scattering was reduced by passing

the beam through a volume holographic beam-splitter and two OD > 4 notch filters (Ondax Inc., USA). Finally, the beam was focused on a 1 m long spectrometer (FHR 1000, Horiba) dispersed by a 600 groove per millimeter grating and detected by a Si CCD (Horiba Inc., USA). Measurements were taken in an ambient environment. Unpolarized spectra were obtained by summing the spectra of two perpendicular incident polarizations, each collected both in parallel and perpendicular configurations. The Raman spectra presented in the main figure are the reduced spectra, which are divided by the Bose-Einstein distribution function to reduce the effect of phonon population for the reasonable comparison with the simulated data from AIMD. The measured frequency range (0–300 cm$^{-1}$) corresponds mainly to lattice vibrations rather than to intra-molecular vibrations of the TPBi CMM. Specifically, the lowest peaks in this range originate almost entirely from the inorganic framework, as molecular peaks appear at higher frequencies[53].

## Interaction analysis of CMMs with perovskite and ligands

XRD patterns of perovskite were obtained by using an X-ray diffractometer (D8-Advance, Bruker). ATR-FTIR spectra of the perovskite films were collected by FT-IR spectrometer (Tensor 27, Bruker). To study the interaction of CMMs with FABr and GABr using $^1$H NMR analysis, FABr and GABr were dissolved in DMSO-d6 (99.9 atom% D, contains 0.03% (v/v) tetramethylsilane (TMS)). CMMs were dissolved in the same solvent with a concentration of 20 mM. The mixture of them was prepared with a molar ratio of 1:1. In order to investigate the interaction between CMMs and ligands, ligand solutions (OA and DAm) were prepared with a concentration of 20.83 mM using benzene-d6 (99.6 atom% D, containing 0.03% (v/v) TMS), respectively. The same solvent was also used to prepare CMM solutions with a concentration of 2 mM. To analyze the interaction between CMMs and ligands, 20 μl of ligand solutions were mixed into 0.5 ml of CMM solutions. $^1$H NMR spectra were obtained by using a 600 MHz NMR spectrometer (Advance 600, Bruker). All $^1$H NMR chemical shifts were acquired at room temperature and calibrated by TMS as an internal reference ($\delta$ = 0 ppm). XPS spectra of bulk perovskite and PeNC films were measured by X-ray photoelectron spectroscope (AXIS-His, KRATOS). UPS analysis was performed using a photoelectron spectrometer (SUPRA, Kratos) combined with a He I radiation source (21.2 eV). SEM images of perovskite film and device were acquired using a scanning electron microscope (SUPRA 55VP, Carl Zeiss AG). TEM images of the PeNCs were obtained using a JEOL JEM-ARM200F microscope, operated at 200 kV and equipped with a K3 IS camera (Gatan, Inc.), installed at the National Center for Inter-university Research Facilities.

## Ellipsometry measurements

Variable angle spectroscopic ellipsometer (J.A. Woollam Co.) was used for measuring the refractive index and thickness of the layers in the PeLED devices. Ellipsometric data was acquired in the range of the wavelength from 200 nm to 1650 nm and the detection angle from 45° to 75° with 5° intervals.

## Optical simulation of PeLED device

Optical simulations were performed with an optical simulation software package (J-OSTD, JooAm Co.) based on the classical dipole model. The refractive indices of the materials comprising the device were obtained from VASE. Optical simulations were performed assuming a perfect charge balance and an evenly distributed emission zone in the perovskite emitting layer. PLQY of 100% and horizontal transition dipole ratio of 66.7% obtained from optical measurements were used as the simulation parameter.

## Data availability

The data that support the findings of this study are available from the corresponding authors upon request.

## Code availability

Code used for calculation of the VDOS is available at https://doi.org/10.5281/zenodo.11497903.

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

## Acknowledgements

This work was supported by the National Research Foundation of Korea (NRF) grant funded by the Ministry of Science and ICT (MSIT) (NRF-2016R1A3B1908431) and NRF grant Brain Link program (2022H1D3A3A01081288). The first-principles DFT modeling of CMM-perovskite interactions with FAPbBr₃ and (FA₁₋ₓGAₓ)PbBr₃, elasticity calculations, and corresponding AIMD simulations for TPBi CMM, and all isolated CMM calculations were conducted by C.P.H. under the support of the NSF Center for Integration of Modern Optoelectronic Materials on Demand (IMOD), an NSF Science and Technology Center (STC) supported by NSF grant DMR-2019444. The first-principles DFT modeling of CMM-perovskite interactions with FAPbBr₃, elasticity calculations, and corresponding AIMD simulations for PO-T2T and 3TPYMB CMMs, were conducted by A.M.S., Z.D., and A.M.R. under support from the U.S. Department of Energy, Office of Science, Basic Energy Sciences, under award no. DE-FG02-07ER46431. Computational support was provided by the National Energy Research Scientific Computing Center (NERSC), a U.S. Department of Energy, Office of Science User Facility located at Lawrence Berkeley National Laboratory, operated under contract no. DE-AC02-05CH11231.

## Author contributions

D.-H.K., S.-J.W., M.-H.P., and T.-W.L. conceived the research idea, initiated the research, and designed the experiments. C.P.H. and A.M.R., in collaboration with Z.D. and A.M.S., conceived the lattice-strengthening idea and designed the theoretical computations. D.-H.K. synthesized colloidal PeNCs. D.-H.K. and S.-J.W. fabricated perovskite nanocrystal films and devices. S.-J.W. and D.-H.K. conducted PL and EL measurements and characterized the luminescence properties of colloidal PeNCs. D.-H.K. and S.-J.W. performed XRD, NMR, and FT-IR measurements and analyzed the interaction between colloidal PeNC and CMMs. C.P.H., A.M.S., and Z.D. performed DFT calculations and AIMD simulations. A.M.S. and Z.D. obtained the VDOS from AIMD simulations. M.-H.P. and J.-M.H. fabricated bulk perovskite films and devices. M.-H.P. performed XRD and SEM measurements for bulk perovskite films. D.-H.K. and J.-M.H. carried out XPS measurements. S. Kim assisted in fabricating colloidal PeNC-LEDs. G.R. and O.Y. carried out Raman spectroscopies. S. Kang and Jungwon Park performed TEM measurements. J.S.K conducted angle-dependent EL measurement and obtained SEM

images of PeNC films and devices. H.J.Y. performed the UPS analysis. D.-H.K. and Jinwoo Park conducted temperature-dependent PL measurements. D.-H.K., S.-J.W., C.P.H., A.M.R., and T.-W.L. primarily wrote the manuscript, and all authors discussed the results and commented on the manuscript. A.M.R. and T.-W.L. supervised the study.

## Competing interests

The authors declare that they have no competing interests.
