## [Peer Review File · Nature Communications]

Surface-Binding Molecular Multipods Strengthen the Halide Perovskite Lattice and Boost LuminescenceREVIEWER COMMENTS

Reviewer #1 (Remarks to the Author):

This work by Kim et al, introduced an interesting concept of using conjugated molecules to stabilize the lattice dynamics for improving electroluminescence of perovskite nanocrystal LEDs. They have systematically investigated a series of n-type molecules and observed enhanced PLQY and longer PL lifetime. An impressive EL EQE was achieved, presenting one of the state-of-the-art. Overall, this is an interesting study that can likely bring in a significant impact to the field, which thus merits publishing in Nature Comm. However, a few claims are not well supported by experimental evidences, and should be carefully considered.

- 1) The lattice stabilization is not validated by experiment. The authors firmly believe the CMMs can stabilize the structure dynamics or ionic fluctuations. But structural evidence is not strong enough. Could the team provide more experimental evidences? Such as FTIR that shows the bond vibration difference, or any other optical spectroscopy measurements that probe the phonon modes? Without probing the suppressed lattice fluctuation, the proposed mechanism is not convincing.
- 2) The Raman measurements shown in Fig. 3h compared the PbBr₆ rotation and FA cation rotation, but more discussion should be added to explain the experiment. What contributes to such motion (laser induced or thermal fluctuation?). And why is octahedron cage rotation reduced but with FA rotation enhanced with TPBi, as the cage rotations are usually related to the cation rotation?
- 3) The authors believe the light induced lattice or ionic fluctuations are important, but this is only the case in the photoluminescence measurements. How is this related to the electroluminescence?
- 4) Molecular interaction between perovskites and CMMs needs more characterizations. The hydrogen bonding and vdW interactions could be probed experimentally by X-ray scattering or IR spectroscopy?
- 5) What would be the difference between thermal evaporated TPBi vs solution coated TPBi? Why do the authors choose to deposit TPBi in the solution phase, as thermally evaporated TPBi could also inter-diffuse to the grain boundary? A discussion could be helpful.
- 6) The operational stability should be studied as this perspective is important for the CMMs engineered PeNC LEDs.

Reviewer #2 (Remarks to the Author):

The manuscript from Dong-Hyeok Kim et al. describes an interesting approach to improve the EQE of perovskite LEDs by mixing commonly employed ETL materials with the perovskite active layer. Similar approaches have been used by other groups on other colloidal nanocrystals (see the work of G. Konstantatos, for example) but they have never been applied to perovskite ones. The approach is relatively innovative, and I guess the manuscript could be of interest for the readership of nature communication. Nonetheless, the manuscript is missing many crucial characterizations of both materials and devices to actually demonstrate that a lattice-strengthening effect is taking place. The modelling carried out by the authors is not sufficient to sustain the various explanations and hypothesis proposed by the authors. I believe much more experimental work is needed before the manuscript can be accepted for publication and to convince the readership that the effects

observed are not due to only a change in the charge balance in the LED and some marginal defect passivation (obtainable with other additives as well). Finally, the addition of ETL materials to Ga-doped FAPbBr NCs seems to have a relatively small effect on the EQE, enough to claim “record efficiency” but far from being a very disruptive finding (champion device EQE of 26% vs roughly 22% for the pristine one).

I attach here below my comments/questions

- Why a volume ratio of 10:1 was selected? Did the authors test various ETL materials content? It would be interesting to see if by changing the ETL materials content the effects on the PL or on the film morphology can be modified/controlled.
- Once again, why the authors selected MAPbBr and FAPbBr NCs? Why different cations were employed for bulk and NCs? Why only these two of the large library of perovskite materials?
- A complete characterization of the perovskites employed is lacking. The authors only focus on the PL characterization in film, while they should present (at the very least):
 - o Optical absorption spectrum of both the ETL materials and the perovskites (bulk and NCs). This is important to understand if any form of energy transfer can take place between the two phases (e.g. TPBi absorbs in the blue spectral region). Similarly optical absorption spectra and discussion of all the composite films is required.
 - o Characterization of the perovskite NCs in solution. Do the ETL materials impact the PL/ABS in solution as well? If they truly induce lattice-strengthening, such effect should be visible in solution too (maybe employing a higher concentration of ETLs)
 - o How were the non-radiative rates estimated? Looking at the TRPL, these are multi-exponential decays (how were they fit?), the authors need to explain how they extract such numbers. Extraction of these values is not as trivial as the authors make it seem, in particular if some resemblance of physical meaning is to be retained.
 - o Do the ETL materials affect the stability of the perovskites? Stability is completely neglected in the manuscript while this is currently one of the most important figure-of-merit for perovskites. Moreover, improved stability is a real fingerprint of increased lattice-strengthening as the soft nature of perovskites is one of the main reasons of their low functional lifetime.
 - o Please include TEM images of the NCs as well, similar to the SEM shown for the MAPbBr film. The authors should also rule out that the improve in PLQY in the NC film is not due to increased spacing between the NCs caused by the ETL additives (i.e., are these NCs mono-disperse? Do poly dispersity change after the ETL addition? Does self-absorption change in the film upon the ETL addition?). Some PL spectra report a small blue-shift (see figure S12) which could indicate a change in self-absorption.
- Regarding the LEDs, many important characterizations are missing in my opinion. First of all, can the authors describe in the experimental method how the EQE was measured? The reported EQE is obtained through an integrating sphere? Or it is referring to only the light emitted in the forward direction?
- The authors completely neglect that the ETL addition can change the energy levels of the active layer. The authors should perform a detailed UPS study of the films for the various compositions employed to understand how the additives are changing the VBM/CBM of the

system. This is to verify how the energy levels are changing and how this could affect charge transport in the device.

- In addition to the UPS, the authors must discuss how the electrical properties of the films with ETL additives are different from the pristine one. Here, we are talking of adding relatively conductive molecules, I would expect some considerable changes in the charge transport properties. Do the films become more n-type? Does the carrier density increase? How can the authors assign the observed effects to lattice-strengthening while they completely disregard the electrical properties? The authors should produce electron and hole only devices and study how the charge transport change.
- An energy band diagram and a device cross-section should be provided.
- Once again, stability is completely disregarded, the authors should provide data for the functional lifetime of their devices and compare it against a pristine reference. Many examples of these measurements are reported in the literature.
- The authors claim to have a nearly perfect device limited only by optical outcoupling. Convincing evidence must be provided, for example the authors must demonstrate that they have nearly reached perfect charge balancing in the active layer.

The manuscript is quite interesting, but it lacks a considerable amount of experimental data to support the lattice-strengthening claimed by the authors. The modelling is not sufficient to sustain their claims in my view.

Reviewer #3 (Remarks to the Author):

In this paper, Kim et al. reports that they improve the EQE of LED device based on perovskite nanocrystals by using conjugated molecular multipods (CMMs) to passivate the nanocrystal surfaces. Although many papers about surface passivation to improve LED efficiency have been reported, surface passivation related to the dynamic lattice disorder in this paper is an original concept. However, there are still some problems should be solved before this paper is published in Nature Communications.

1、 The authors claim that hole transporting molecules would result in exciplex formation in EL devices (Supplementary Fig. 1-3). How to rule out the parasitic emission from the perovskite layer and not the transport layer? In addition, this discussion is based on the bulk perovskite materials, how about perovskite nanocrystals? In perovskite nanocrystals case, is there direct evidence for parasitic emission in EL devices when conjugated hole transporting molecules passivate nanocrystal surface?

2、 It can be seen that the highest EQE are shown at 3.4 V, which the current density is lower than 0.5 mA cm⁻¹. Due to the current density value is too little, EQE measured by integrated sphere method is required to make sure the data are right. By the way, the angular emission profile of the device should be offered to prove the LED is a Lambertian emitter if the authors use a spectroradiometer to measure the EQE. Otherwise, the EQE data should be corrected according the angular emission profile.

3、 What are the thicknesses of GraHIL and GA-doped perovskite nanocrystal EML? The cross-section transmission electron microscopy (TEM) image of best LED device is required to verify that the parameters of the optical simulation are correct.

4、 The authors list a table of state-art-of green PeNC LEDs performance in Table S1. In this table, the device in this work is the best performance. In fact, EQEs of green PeNC LEDs over 25% have been reported by others (Nature 2022, 611, 688-694; Adv. Mater. 2023, 2302283; ACS Energy Lett. 2023,8, 927-934.). The authors should also list these data in the Table S1, otherwise, they risk misleading the readers that their work is the best result so far.

5、 How is the stability of PeNC LED with high EQE (26.1%)? The authors do not demonstrate the result. Is the stability not good?

6、 The TEM images of perovskite nanocrystals are suggested to offered by the autho

Response to the Reviewers' Comments

[Nature Communications NCOMMS-23-31521-T]: Surface-Binding Molecular Multipods Strengthen the Halide Perovskite Lattice and Boost Luminescence

We would like to thank the reviewers and the editors for their valuable comments on our work. Such productive feedbacks from the world experts in this field helped us improve our manuscript. We provide a point by point response below to the reviewers' questions. We also revised our manuscript to comply with the reviewers' comments and have highlighted the revised parts in red.

Reviewer #1 (Remarks to the Author):

This work by Kim et al, introduced an interesting concept of using conjugated molecules to stabilize the lattice dynamics for improving electroluminescence of perovskite nanocrystal LEDs. They have systematically investigated a series of n-type molecules and observed enhanced PLQY and longer PL lifetime. An impressive EL EQE was achieved, presenting one of the state-of-the-art. Overall, this is an interesting study that can likely bring in a significant impact to the field, which thus merits publishing in *Nature Comm*. However, a few claims are not well supported by experimental evidences, and should be carefully considered.

1) The lattice stabilization is not validated by experiment. The authors firmly believe the CMMs can stabilize the structure dynamics or ionic fluctuations. But structural evidence is not strong enough. Could the team provide more experimental evidences? Such as FTIR that shows the bond vibration difference, or any other optical spectroscopy measurements that probe the phonon modes? Without probing the suppressed lattice fluctuation, the proposed mechanism is not convincing.

Reply: Thank you very much for the valuable comment. Since Raman measurement is the most direct evidence to prove the lattice strengthening effect, we planned to perform additional Raman measurements with our professional collaborators at the Weizmann Institute (Israel). Our collaborator's Raman system is carefully designed to analyze low-frequency modes under 300 cm^{-1} which is not suitable for typical commercial Raman spectrometers. Our collaborator's Raman system correctly calibrates the low-frequency Raman signal of perovskite by removing the Rayleigh scattering of the laser source. However, due to the sudden situation in Israel during the revision period, we were not able to do additional Raman measurements.

Despite this situation, we comprehensively investigated the interaction between CMMs and perovskite in order to prove our claim for the improvement of luminescent efficiency by lattice-strengthening effect. Through $^1\text{H-NMR}$ and FT-IR analysis, we identified the formation of the

hydrogen bonding between CMMs and perovskite, which is the fundamental mechanism for lattice-strengthening (Fig. 4d,e and Fig. 5a). In $^1\text{H-NMR}$ spectra, downfield shift in $-\text{NH}_2$ signals of both FA and GA cations clearly indicated the formation of the hydrogen bonding between CMMs and cations (Fig. 4d and Fig. 5a). Furthermore, ATR-FTIR spectra of FAPbBr_3 and $\text{FAPbBr}_3\text{-TPBi}$ showed decreased wavenumber and peak broadening of N-H stretch mode in FA cations, revealing the presence of the hydrogen bonding at the surface of perovskite crystal (Fig. 4e). Both results clearly demonstrated that CMMs strengthen the perovskite lattice by forming hydrogen bonding with cations of perovskites.

We additionally measured XRD, TEM, SEM, and UPS to prove that crystal structure, particle size, morphology, valence band maximum (VBM) and conduction band maximum (CBM) of the PeNCs were not affected by adding CMMs. First, we measured XRD patterns of perovskite films to study whether the CMMs affect the crystal structure of the perovskite (Fig. 4a and Supplementary Figs.11,19). There were no peak shifts in XRD patterns, indicating that CMMs do not change the crystal structure of perovskite. Furthermore, through TEM measurement, we identified that the average size and particle size distribution of colloidal PeNCs were not affected by CMM incorporation (Fig. 4b and Supplementary Figs.12,21). SEM images of perovskite films also showed that the change in the film morphology has less relevance with luminescent efficiency (Supplementary Figs.13,14,20). Furthermore, combined analysis of UPS spectra and UV-vis absorption spectra demonstrated that CMM does not affect the VBM and CBM of PeNCs (Fig. 4c and Supplementary Figs. 15,16).

In conclusion, we are confident that the lattice-strengthening effect improved luminescent performance. From the new $^1\text{H-NMR}$ and FT-IR analysis, we clearly proved the hydrogen bonding formation between CMMs and cations. Our experimentally measured Raman spectra showed increased vibrational frequencies of PbBr_6 octahedral distortion/rotation by TPBi CMM incorporation (Fig. 3h), which are supported by the theoretically calculated Raman spectra showing that all CMMs suppress low-frequency dynamics (Fig. 3g and Supplementary Fig. 10). Consistent results from experiments and theoretical calculations of Raman spectra clearly showed the lattice-strengthening effect by CMMs. Calculated elastic modulus of perovskite well explained the luminescent characteristics of both bulk perovskite and PeNCs (Fig. 2,3,5). Furthermore, other possible effects of CMM addition such as their effect on crystal structure, particle size, and energy band of PeNCs were negligible (Fig. 4). Overall, these results ensure that the lattice-strengthening mechanism enhances the luminescence of perovskite.

Revision: Page 8 line 13 in the revised manuscript, Reference, Fig. 4, Supplementary Figs. 11-16, Page 9 line 27 in the revised manuscript, Fig. 5a, Supplementary Figs. 19-21, and their explanations

Interaction between perovskites and CMMs

To evaluate alternative explanations for the improved luminescent properties, we studied whether CMM incorporation has additional effects beyond strengthening the near-surface lattice. To determine whether CMMs influence the crystal structure of perovskites, we conducted X-ray diffraction (XRD) measurement (Fig. 4a and Supplementary Fig. 11). The FAPbBr₃ PeNCs with CMMs showed the same XRD patterns that correspond to the cubic phase²⁰, indicating that the incorporation of CMMs did not induce any change in the crystal structures. High-resolution transmission electron microscopy (TEM) images of colloidal FAPbBr₃ PeNCs showed average particle size of ~9.78 nm without CMM, 9.90 nm with 3TPYMB, 9.98 nm with PO-T2T, and 9.88 nm with TPBi (Fig. 4b and Supplementary Fig. 12). This result indicates that post-synthesis blending of CMMs into colloidal PeNCs has negligible effect on the average particle size and distribution. Furthermore, CMM incorporations caused no significant change in the morphology of perovskite films (Supplementary Figs. 13,14). Combined analysis of ultraviolet photoelectron spectroscopy (UPS) spectra and UV-vis absorption spectra confirmed that all CMMs do not affect the valence band maximum (VBM) or conduction band maximum (CBM) of PeNCs (Fig. 4c and Supplementary Figs. 15,16). The work functions were not changed by CMMs of TPBi and 3TPYMB, but the incorporation of PO-T2T CMM lowered the work function by 0.17 eV, making FAPbBr₃ PeNC with PO-T2T slightly more n-type than the FAPbBr₃ PeNC without CMM. However, the effect of CMMs on the current density of the electron-only device was negligible despite the electron-transporting nature of the CMMs (Supplementary Fig. 17). On the other hand, due to the hole-blocking characteristics of the CMMs, the current density of the hole-only device was largely decreased (Supplementary Fig. 17). This led to the decreased current density of colloidal FAPbBr₃ PeNC-LEDs with PO-T2T and TPBi CMMs compared to that of the device without CMM. Thus, as the CMMs do not significantly impact the crystal structure, particle size distribution, film morphology, VBM, and CBM of PeNCs, the lattice strengthening effect is the primary contributor to the improved luminescent efficiency.

To experimentally confirm the mode of interaction between the CMMs and the perovskite, a solution nuclear magnetic resonance (NMR) analysis was conducted using FABr, dissolved with PO-T2T or TPBi in DMSO-d₆ (Fig. 4d). The -NH₂ signals in FABr were obviously shifted downfield when TPBi or PO-T2T were included. This result indicates formation of hydrogen bonds between FA cation and the nucleophilic functional groups that have lone pair electrons (benzimidazole and phosphine oxide group) in the CMMs^{42,43}. To confirm this effect between the CMMs and the FA cations at the surface of the perovskite crystal lattice, we measured attenuated total reflectance-Fourier transform infrared (ATR-FTIR) spectra of FAPbBr₃ and FAPbBr₃-TPBi (Fig. 4e). Addition of TPBi caused decrease in wavenumber and peak broadening of both the symmetric N-H stretching mode ($\nu_s(\text{N-H})$) and the C=N stretching mode ($\nu(\text{C=N})$); these changes clearly shows the hydrogen bonding formation between FA cation and TPBi^{43,44}. X-ray photoelectron spectroscopy (XPS) analysis of Br 3d states in CMM-FAPbBr₃ mixed films shows no evidence of strong interaction between CMMs and the surface Br ions (Supplementary Fig. 18). Together, these results confirm our theoretical predictions and demonstrate that CMMs are adsorbed to the surface of the perovskite nanocrystals by hydrogen bonding with FA cations.

Reference

42. Li, N. *et al.* Cation and anion immobilization through chemical bonding enhancement with fluorides for stable halide perovskite solar cells. *Nat. Energy* **4**, 408–415 (2019).
43. Xu, Y. *et al.* Perovskite Films Regulation via Hydrogen-Bonded Polymer Network for Efficient and Stable Perovskite Solar Cells. *Angew. Chemie Int. Ed.* **62**, e202306229 (2023).
44. Xu, W. *et al.* Rational molecular passivation for high-performance perovskite light-emitting diodes. *Nat. Photonics* **13**, 418–424 (2019).

Fig. 4 | Interaction between FAPbBr₃ PeNC and CMMs. **a-c**, Independent characteristics of FAPbBr₃ PeNCs with CMMs. Crystal structure (**a**) of FAPbBr₃ PeNCs. X-ray diffraction (XRD) patterns of FAPbBr₃ PeNC films. Incorporation of CMMs did not cause a peak shift. The size distribution (**b**) of colloidal FAPbBr₃ PeNCs. Particle size was extracted from transmission electron microscope (TEM) images. Electronic structure (**c**) of FAPbBr₃ PeNCs. Energy band diagrams of FAPbBr₃ PeNCs with CMMs were obtained by combining ultraviolet photoelectron spectroscopy (UPS) analysis and optical bandgap. Valence band maximum (VBM) and optical bandgap were derived from UPS spectra and UV-vis absorption spectra, respectively. The conduction band maximum (CBM) was calculated based on the VBM level and optical bandgap of PeNCs. **d-e**, Hydrogen bonding formation between CMMs and perovskite. The ¹H-NMR spectra (**d**) of FABr and mixtures of FABr, TPBi and PO-T2T. -NH₂ signal of FABr marked with violet color on FABr molecule. The ¹H-NMR spectrum of pure PO-T2T was shown in the dashed line. Excluding the peaks arising from PO-T2T, the peak shift of the -NH₂ signal in FA can be identified. Hydrogen bonding between FABr and CMMs was indicated by a green dashed line. ATR-FTIR spectra (**e**) of FAPbBr₃ and FAPbBr₃-TPBi without ligands. The region of symmetric N-H stretching mode $\nu_s(\text{N-H})$ and C=N stretching mode $\nu(\text{C=N})$ was magnified in the graph below.

Supplementary Figs. 11-16, and their explanations:

Figure S11 | XRD patterns of bulk MAPbBr₃ with CMMs.

XRD patterns of bulk MAPbBr₃ with CMMs were measured to identify the effect of CMMs on crystal structure. Bulk MAPbBr₃ film with no CMM showed peaks at 14.98°, 21.26°, 30.26°, 33.9°, 37.24°, 43.28°, and 46.04°, which can be assigned (100), (110), (200), (210), (211), (220) and (300) planes, respectively. These peak positions correspond to the cubic phase of MAPbBr₃⁴. Furthermore, there was no peak shift when CMMs were incorporated into bulk MAPbBr₃, indicating that CMMs do not affect the crystal structure of bulk MAPbBr₃.

Figure S12 | TEM images of colloidal FAPbBr₃ PeNCs with CMMs. a, no CMM, b, 3TPYMB, c, PO-T2T and d, TPBi. Each image size is 40 nm × 40 nm.

Figure S13 | Scanning electron microscope (SEM) images of FAPbBr₃ PeNC films embedding CMMs.

Figure S14 | SEM images of bulk MAPbBr₃ films embedding CMMs with grain size distribution. a, no CMM, b, 3TPYMB, c, PO-T2T and d, TPBi.

To investigate the morphological effect of CMMs on bulk MAPbBr₃, we measured SEM images of CMM-embedded bulk MAPbBr₃ films. Bulk MAPbBr₃ with no CMM showed an average grain size of 159.34 nm. Because CMMs are embedded into perovskite during the crystallization process (additive-based nanocrystal pinning effect)⁴, the incorporation of 3TPYMB, PO-T2T, and TPBi reduced average grain size to 86.13 nm, 96.24 nm, and 84.04 nm, respectively, which are very similar. We could not find a clear correlation between morphology and luminescent efficiency of bulk MAPbBr₃ upon incorporation of various CMM (Fig. 2 and Supplementary Fig. 8d-f).

Figure S15 | UPS spectra of FAPbBr₃ PeNC films with CMMs.

Figure S16 | UV-vis absorption spectra. a, CMM films, b-d, Perovskite films with CMMs. bulk MAPbBr₃ (b), FAPbBr₃ PeNCs (c), FA_{0.9}GA_{0.1}PbBr₃ PeNCs (d).

Solution $^1\text{H-NMR}$ analysis showed a downfield shift of the $-\text{NH}_2$ signals in GA cations blended with TPBi or PO-T2T, indicating that GA cation also form hydrogen bonding with CMMs (Fig. 5a). Similar to the studies of FAPbBr_3 PeNCs (Fig. 4a,b and Supplementary Figs. 12,13), CMM incorporation had negligible impact on the crystal structure, film morphology, and particle size of GA-doped FAPbBr_3 PeNCs (Supplementary Figs.19-21).

Fig. 5 | Effect of CMMs on $\text{FA}_{0.9}\text{GA}_{0.1}\text{PbBr}_3$ PeNCs and their luminescence characteristics. a, $^1\text{H-NMR}$ spectra of GABr and mixtures of GABr, TPBi, and PO-T2T. $-\text{NH}_2$ signal of GABr was marked with green color on the GABr molecule.

Figure S19 | XRD patterns of $\text{FA}_{0.9}\text{GA}_{0.1}\text{PbBr}_3$ PeNCs films with CMMs.

Figure S20 | SEM images of $\text{FA}_{0.9}\text{GA}_{0.1}\text{PbBr}_3$ films embedding CMMs.

Figure S21 | TEM images of colloidal $\text{FA}_{0.9}\text{GA}_{0.1}\text{PbBr}_3$ PeNCs with CMMs. **a, no CMM, **b**, PO-T2T, **c**, TPBi, and **d**, size distributions. Each image size is 40 nm \times 40 nm.**

2) The Raman measurements shown in Fig. 3h compared the PbBr_6 rotation and FA cation rotation, but more discussion should be added to explain the experiment. What contributes to such motion (laser induced or thermal fluctuation?). And why is octahedron cage rotation reduced but with FA rotation enhanced with TPBi, as the cage rotations are usually related to the cation rotation?

Reply: We thank the reviewer for pointing out the ambiguity in our discussion of these experimental and theoretical results. We have significantly revised the discussion of the mechanism in the section “Effect of conjugated molecular multipods on perovskite lattice dynamics” to clarify the relationship between the different experimental and computational techniques and the lattice strengthening mechanism. As has been previously demonstrated both theoretically and through Raman measurements, at room temperature the low-frequency cage distortion modes have significant populations [*Phys. Rev. Lett.* **118**, 136001 (2017)]. These large amplitude thermal motions have important effects on carrier transport and recombination [*J. Phys. Chem. Lett.* **6**, 4754-4757 (2015) / *Nano Lett.* **18**, 8041-8046 (2018)], so we anticipate that changes to these vibrations will impact device performance.

Using strain calculations of model systems, we predict that the multi-site anchoring of the CMMs will stiffen the near-surface region of the perovskite. As stiffer bonds correspond to higher vibrational frequencies, we predict an overall blue shift of the low-frequency vibrations. Indeed, we observed such a shift in both MD predictions and Raman measurements, which is clear evidence of lattice-strengthening. Crucially, we do not predict a decrease in vibrational populations, and indeed the Raman spectra are not normalized in a way that permits comparison of peak heights between samples. However, an increase in the frequency of the $\sim 50 \text{ cm}^{-1}$ modes in the CMM-containing sample is clearly observed, corresponding to a stiffening of the octahedral cage modes. The referee’s point that cage dynamics is intimately linked with cation motion is well taken. We have clarified that CMMs coordinate with the surface of perovskite by FA cations, a fact which is supported by theoretical modeling and new experiments. Despite this, only a small fraction of surface FA cations are bound to the CMM, while the rest of the surface cations are not directly affected, as they can rotate freely within the lattice. On the other hand, as the PbBr lattice is directly connected, the anchoring likely has a more uniform effect across the surface. This can explain why any peak shifting in the Raman spectrum is much less prominent compared to the shifting and broadening in the first peak which is associated with octahedral rotation. Therefore, we focus on the Pb octahedral cage modes to verify lattice-strengthening, and we removed labels indicating FA cation rotation/translation in Fig. 3h to focus on the part of the signal most clearly associated with lattice-strengthening.

Revision: Fig. 3h

Before:

Fig. 3 | Lattice-strengthened FAPbBr₃ perovskite by CMMs. h, Experimentally-measured low-frequency Raman spectra of FAPbBr₃ PeNC films with and without TPBi.

After:

Fig. 3 | Lattice-strengthened FAPbBr₃ perovskite by CMMs. h, Experimentally-measured low-frequency Raman spectra of FAPbBr₃ PeNC films with and without TPBi.

3) The authors believe the light induced lattice or ionic fluctuations are important, but this is only the case in the photoluminescence measurements. How is this related to the electroluminescence?

Reply: Thank you very much detailed question. In the manuscript, we are correlating the inherent dynamic disorder of perovskites due to their weak ionic bonding characteristics (*PRL* **118**, 136001 (2017)) with their luminescence properties (either for PL and EL). Based on the experimental and theoretical calculations, we found that CMMs bind with the perovskite surface via hydrogen bonding which led to the increase in both the photoluminescence and electroluminescence efficiency.

We performed large scale, finite temperature *ab initio* molecular dynamics of combined

perovskite-CMM model systems. The significant computational expense of these simulations enables us to accurately simulate the anharmonic room temperature motions of the perovskite lattice and predict a slight stiffening of the low frequency vibrational modes. However, our modeling included only thermal vibrational populations and did not consider additional vibrational excitation due to optical probes. Our experimentally measured Raman corroborated this finding, showing a similar blue shift in the low-frequency peak of the vibrational spectrum, which similarly involved no electronic excitation.

In conclusion, we incorporated CMMs to perovskite to strengthen the perovskite lattice and suppress intrinsic dynamic disorder. By virtues of both experimental measurement and theoretical calculation, we confirmed that CMMs suppressed the inherent dynamic disorder in the perovskite even without external stimulation (Fig. 3). This is an approach to suppress the non-radiative pathway existing in the materials itself, thus is effective not only to PL but also to EL performance (Fig. 2,5).

4) Molecular interaction between perovskites and CMMs needs more characterizations. The hydrogen bonding and vdW interactions could be probed experimentally by X-ray scattering or IR spectroscopy?

Reply: We appreciate the reviewer for asking this important question. All atoms or molecules that are close to each other to form a vdW force, which is highly sensitive to the distance and also their conformation. Our theoretical calculation considered vdW force to obtain the elastic modulus of perovskites with adjusting molecular conformation and distance (Fig. 3), but it is technically challenging to experimentally quantify vdW force between the perovskite crystal and CMMs. However, we clearly confirmed the hydrogen bonding between CMMs and perovskite by experimental $^1\text{H-NMR}$ and ATR-FTIR measurements, which is shown in Fig. 4d,e and Fig. 5a. To verify the presence of the hydrogen bonding formation between cations of perovskite and CMMs, we dissolved CMMs (TPBi and PO-T2T) and cations (FA and GA) in the solvent (DMSO- d_6). The CMM incorporation induced an obvious downfield shift of $-\text{NH}_2$ signals in both FA and GA cations. This is because the hydrogen bonding formation between the ammonium group in cations and lone pair electrons in CMMs makes a deshielding of NH_2 protons in cations. This result is consistent with the previous studies conducting $^1\text{H-NMR}$ analysis to prove hydrogen bonding (*Nat. Energy* **4**, 408–415 (2019) / *Angew. Chemie Int. Ed.* **62**, e202306229 (2023)). To elucidate the formation of the hydrogen bonding between the CMMs and FA cations at the surface of the perovskite crystal lattice, we additionally measured ATR-FT-IR spectra of FAPbBr_3 and $\text{FAPbBr}_3\text{-TPBi}$. Symmetric N-H stretch mode ($\nu_s(\text{N-H})$) showed a decrease in wavenumber and peak broadening, which is direct evidence of hydrogen bonding. Our results are also matched with the reports utilizing the FT-IR technique

to study hydrogen bonding (*Angew. Chemie Int. Ed.* **62**, e202306229 (2023) / *Nat. Photonics* **13**, 418–424 (2019)). Additionally, the C-N vibrational mode also showed a decrease in wavenumber and peak broadening, consistently demonstrating that hydrogen bonding is formed between CMMs and cations at the surface of perovskite crystal.

Revision: Page 9 line 8 in the revised manuscript, Reference, Fig. 4d-e, Page 9 line 27 in the revised manuscript, and Fig. 5a.

Page 9 line 8 in the revised manuscript, Reference, and Fig. 3d-e:

To experimentally confirm the mode of interaction between the CMMs and the perovskite, a solution nuclear magnetic resonance (NMR) analysis was conducted using FABr, dissolved with PO-T2T or TPBi in DMSO-d₆ (Fig. 4d). The -NH₂ signals in FABr were obviously shifted downfield when TPBi or PO-T2T were included. This result indicates formation of hydrogen bonds between FA cation and the nucleophilic functional groups that have lone pair electrons (benzimidazole and phosphine oxide group) in the CMMs^{42,43}. To confirm this effect between the CMMs and the FA cations at the surface of the perovskite crystal lattice, we measured attenuated total reflectance-Fourier transform infrared (ATR-FTIR) spectra of FAPbBr₃ and FAPbBr₃-TPBi (Fig. 4e). Addition of TPBi caused decrease in wavenumber and peak broadening of both the symmetric N-H stretching mode ($\nu_s(\text{N-H})$) and the C=N stretching mode ($\nu(\text{C=N})$); these changes clearly shows the hydrogen bonding formation between FA cation and TPBi^{43,44}. X-ray photoelectron spectroscopy (XPS) analysis of Br 3d states in CMM-FAPbBr₃ mixed films shows no evidence of strong interaction between CMMs and the surface Br ions (Supplementary Fig. 18). Together, these results confirm our theoretical predictions and demonstrate that CMMs are adsorbed to the surface of the perovskite nanocrystals by hydrogen bonding with FA cations.

Reference

42. Li, N. *et al.* Cation and anion immobilization through chemical bonding enhancement with fluorides for stable halide perovskite solar cells. *Nat. Energy* **4**, 408–415 (2019).
43. Xu, Y. *et al.* Perovskite Films Regulation via Hydrogen-Bonded Polymer Network for Efficient and Stable Perovskite Solar Cells. *Angew. Chemie Int. Ed.* **62**, e202306229 (2023).
44. Xu, W. *et al.* Rational molecular passivation for high-performance perovskite light-emitting diodes. *Nat. Photonics* **13**, 418–424 (2019).

Fig. 4 | Interaction between FAPbBr₃ PeNC and CMMs. d-e, Hydrogen bonding formation between CMMs and perovskite. The ¹H-NMR spectra (d) of FABr and mixtures of FABr, TPBi and PO-T2T. –NH₂ signal of FABr marked with violet color on FABr molecule. The ¹H-NMR spectrum of pure PO-T2T was shown in the dashed line. Excluding the peaks arising from PO-T2T, the peak shift of the –NH₂ signal in FA can be identified. Hydrogen bonding between FABr and CMMs was indicated by a green dashed line. ATR-FTIR spectra (e) of FAPbBr₃ and FAPbBr₃-TPBi without ligands. The region of symmetric N-H stretching mode $\nu_s(\text{N-H})$ and C=N stretching mode $\nu(\text{C=N})$ was magnified in the graph below.

Page 9 line 27 in the revised manuscript and Fig. 5a:

Solution ¹H-NMR analysis showed a downfield shift of the -NH₂ signals in GA cations blended with TPBi or PO-T2T, indicating that GA cation also form hydrogen bonding with CMMs (Fig. 5a).

Fig. 5 | Effect of CMMs on FA_{0.9}GA_{0.1}PbBr₃ PeNCs and their luminescence characteristics. a, ¹H-NMR spectra of GABr and mixtures of GABr, TPBi, and PO-T2T. –NH₂ signal of GABr was marked with green color on the GABr molecule.

5) What would be the difference between thermal evaporated TPBi vs solution coated TPBi? Why do the authors choose to deposit TPBi in the solution phase, as thermally evaporated TPBi could also inter-diffuse to the grain boundary? A discussion could be helpful.

Reply: Thank you very much detailed question about the fundamental approach. As shown in Fig.1 schematic, we blended PeNC and CMM solution before the spin coating process. This approach allows TPBi to be distributed homogeneously across the film. However, treating TPBi by thermal evaporation affects the surface of PeNCs only at the top of the film. Furthermore, the lateral size of the CMM molecule matches the 3 to 4-unit cell of the perovskite lattice (Fig. 3a), making it difficult to diffuse inside the film along the grain boundary or interparticle space during the evaporation process. This will limit the perovskite surface coverage by CMMs, resulting in a small lattice strengthening effect. Therefore, to maximize the surface coverage and lattice-strengthening effect, we utilized solution phase blending.

To further show that solution-phase blending has a superior lattice strengthening effect over the evaporation process, we compared the PLQY and transient PL curves of $\text{FA}_{0.9}\text{GA}_{0.1}\text{PbBr}_3$ with no CMM and thermally evaporated TPBi. As can be seen below, we found that thermally-evaporated TPBi on the PeNC film leads to decreased PLQY (Fig. R1). We believe that evaporated TPBi on top of the PeNC films induced other side effects leading to degraded luminescent performance. We postulate the decrease in luminescent performance to two possible reasons: 1) Thermal degradation during TPBi evaporation. 2) Spontaneous orientation polarization (SOP) of TPBi films. As previously reported, PeNCs are hugely vulnerable to the heat due to ligand detachment and aggregation (*Nature Photonics* **15**, 148-155 (2021) / *Phys. Chem. Chem. Phys.* **19**, 8934-8940 (2017) / *ACS Appl. Mater. Interfaces* **10**, 5984-5991, (2018)). When sublimated high-temperature-TPBi molecules are deposited onto the PeNC film during thermal evaporation, thermal degradation such as ligand detachment or PeNC aggregation can occur. Another possible reason for the decrease in PL performance is SOP of thermally deposited TPBi. Organic molecules that have preferred orientation during thermal deposition leads to the formation of large surface potential due to their intrinsic permanent dipole moment, which is SOP (*Nature Materials* **21**, 819-825 (2022) / *Sci. Adv.* **6**, eabb2659 (2020)). SOP could induce charge accumulation at the ETL/EML interface, leading to exciton quenching (*Sci. Adv.* **6**, eabb2659 (2020)). However, this effect has not yet been actively studied in the field of perovskite and could be further investigated in future research.

Fig. R1. Transient PL decay and PLQY of FA_{0.9}GA_{0.1}PbBr₃ PeNCs with evaporated TPBi.

6) The operational stability should be studied as this perspective is important for the CMMs engineered PeNC LEDs.

Reply: Thank you for the valuable comment. We measured the operational lifetime of colloidal FA_{0.9}GA_{0.1}PbBr₃ PeNC-LED with CMMs at an initial luminance (L_0) of 100 cd/m², as shown in Supplementary Fig. 31. The device with no CMM showed device half-lifetime (LT_{50}) of 1.6 h, but TPBi CMM showed improved device lifetime with LT_{50} of 5h. TPBi CMM also enhanced luminescent efficiency by lattice-strengthening without ligand detachment effect (Fig. 5). On the contrary, PO-T2T showed shorter lifetime (LT_{50} of 0.7h). This may be attributed to the accelerated degradation process during the device operation by ligand detachment effect of PO-T2T having strong reactive characteristics with organic ligand (oleic acid) (Fig. 6). Therefore, we demonstrated that the lattice-strengthening strategy by TPBi that does not induce ligand detachment not only enhances luminescence efficiency but also improves operational stability.

Revision: Page 10 line 26 in the revised manuscript, Supplementary Fig. 31, and its explanation

Page 10 line 26 in the revised manuscript:

In addition to the improvement of efficiency, the TPBi CMM device showed a longer device half-lifetime by a factor of 3.1 compared to the device without CMM, indicating that lattice-strengthening by TPBi that does not induce ligand detachment also contributes to suppressing the degradation process of the perovskite (Supplementary Fig. 31).

Supplementary Fig. 31, and its explanation:

Figure S31 | Operational lifetime of colloidal FA_{0.9}GA_{0.1}PbBr₃ PeNC devices with CMMs.

To investigate the effect of CMM on the stability of colloidal PeNC-LEDs, we measured the device half-lifetime (LT_{50}) of colloidal FA_{0.9}GA_{0.1}PbBr₃ PeNC-LEDs under the condition that the initial luminance was 100 cd/m². Compared to no CMM device (LT_{50} of 1.6h), the TPBi CMM device showed improved device lifetime (LT_{50} of 5h). On the other hand, PO-T2T exhibited reduced device stability (LT_{50} of 0.7h) because it has highly reactive characteristics with oleic acid ligand (Fig. 6). Therefore, we demonstrated that lattice-strengthening by TPBi CMM that does not induce ligand detachment not only enhances luminescent efficiency, but suppresses the degradation of perovskite.

Reviewer #2 (Remarks to the Author):

The manuscript from Dong-Hyeok Kim et al. describes an interesting approach to improve the EQE of perovskite LEDs by mixing commonly employed ETL materials with the perovskite active layer. Similar approaches have been used by other groups on other colloidal nanocrystals (see the work of G. Konstantatos, for example) but they have never being applied to perovskite ones. The approach is relatively innovative, and I guess the manuscript could be of interest for the readership of nature communication. Nonetheless, the manuscript is missing many crucial characterizations of both materials and devices to actually demonstrate that a lattice-strengthening effect is taking place. The modelling carried out by the authors is not sufficient to sustain the various explanations and hypothesis proposed by the authors. I believe much more experimental work is needed before the manuscript can be accepted for publication and to convince the readership that the effects observed are not due to only a change in the charge balance in the LED and some marginal defect passivation (obtainable with other additives as well). Finally, the addition of ETL materials to Ga-doped FAPbBr NCs seems to have a relatively small effect on the EQE, enough to claim “record efficiency” but far from being a very disruptive finding (champion device EQE of 26% vs roughly 22% for the pristine one).

I attach here below my comments/questions

1) Why a volume ratio of 10:1 was selected? Did the authors test various ETL materials content? It would be interesting to see if by changing the ETL materials content the effects on the PL or on the film morphology can be modified/controlled.

Reply: Thank you for the valuable questions. To simplify the solution process of lattice-strengthening in the PeNC-CMM system, we separately prepared PeNC and CMM solution, and blended them before the spin-coating process (Fig. 1). Through the experiment of optimizing TPBi content, we confirmed that the volume ratio of the PeNC solution and TPBi solution (5mM) used as 10:1 was a well-designed condition due to the following reasons. (1) TPBi precursor was well dissolved in the solvent when the concentration of pure TPBi solution was 5mM. (2) The concentration of PeNC did not decrease significantly while containing a sufficient content of TPBi for lattice-strengthening in a PeNC/TPBi mixture.

To verify the lattice-strengthening effect according to the TPBi content, we measured PLQY of PeNC-TPBi films depending on the concentration of pure TPBi solution, as shown in Supplementary Fig. 24a. With increasing TPBi contents, we verified that film PLQY was also improved, indicating increased lattice-strengthening effect. A TPBi concentration of 5 mM showed the best PL performance with near-unity PLQY. However, 7mM TPBi concentration

showed decreased PLQY, which could be attributed to the ligand-detachment effect. Since a sufficient amount of TPBi required for efficient lattice-hardening has already been achieved at 5mM, we assumed that using an excess amount of CMM could induce ligand detachment effect so that reducing the PLQY.

To investigate the effect of CMM on the morphology of PeNC films, we measured top SEM images of FAPbBr_3 and $\text{FA}_{0.9}\text{GA}_{0.1}\text{PbBr}_3$ PeNC films with CMMs, as shown in Supplementary Fig. 13, 20. All conditions showed uniform film coverage without pinholes, and CMM incorporation does not significantly affect film morphology. This indicates that enhanced luminescent properties are less relevant to morphological effect.

Revision: Supplementary Fig. 24a, and its explanation. Page 8 line 24 in the revised manuscript, Supplementary Fig. 13, Page 9 line 29 in the revised manuscript, Supplementary Fig. 20

Supplementary Fig. 24a and its explanation:

Figure S24 | PLQY of $\text{FA}_{0.9}\text{GA}_{0.1}\text{PbBr}_3$ PeNC films depending on the concentration of CMM solutions. a, TPBi CMM.

To investigate how the content of CMMs influences PLQY, we adjusted the concentration of CMM solutions. As the concentration of TPBi solution increased, the effect of lattice-strengthening increased. The PLQY was highest (near-unity) when TPBi solution was 5 mM.

Page 8 line 24 in the revised manuscript and Supplementary Fig. 13:

Furthermore, CMM incorporations caused no significant change in the morphology of perovskite films (Supplementary Figs. 13,14).

Figure S13 | Scanning electron microscope (SEM) images of FAPbBr₃ PeNC films embedding CMMs.

Page 9 line 29 in the revised manuscript and Supplementary Fig. 20:

Similar to the studies of FAPbBr₃ PeNCs (Fig. 4a,b and Supplementary Figs. 12,13), CMM incorporation had negligible impact on the crystal structure, film morphology, and particle size of GA-doped FAPbBr₃ PeNCs (Supplementary Figs.19-21).

Figure S20 | SEM images of FA_{0.9}GA_{0.1}PbBr₃ PeNC films embedding CMMs.

2) Once again, why the authors selected MAPbBr and FAPbBr NCs? Why different cations were employed for bulk and NCs? Why only these two of the large library of perovskite materials?

Reply: Thank you very much for the valuable question. As the reviewer mentioned, lead halide perovskite materials (APbX₃) have a large library according to their A-site cations (A) and halide anions (X). In terms of display application, the color space of Rec. 2020 standard for next-generation vivid displays requires substantially wide coverage in the green color. Thus, it is crucial to develop a high-efficiency green emitter having an emission wavelength near 532 nm (green primary color in Rec. 2020 standard) with a narrow full width at half maximum (FWHM). Although the optical bandgap is mainly formed by lead (Pb) and halide (X) orbital

overlapping, the size of A-site cations also affects the optical bandgap by perovskite lattice contraction or expansion (*APL Mater.* **9**, 109202 (2021)). Therefore, the small size of the Cs cation (1.81 Å) makes the bandgap of CsPbBr₃ larger so that CsPbBr₃ emitters have emission wavelength under 520 nm (*Nano Lett.* **14**, 3608-3616 (2014) / *ACS Appl. Mater. Interfaces* **12**, 2835-2841 (2020)), which hugely deviated from the wavelength of green primary color (532 nm) in Rec 2020. standard. In contrast to the Cs cation, the relatively larger size of FA cation (2.79 Å) and MA (2.70 Å) cations makes the optical bandgap smaller. Thus, our confined FAPbBr₃ PeNCs showed emission peaks at the wavelength of 532 nm, which is the same wavelength as the green primary color in the Rec. 2020 standard. That is why we adopted to use organic cations in our studies. In addition, in order to further improve the luminescence efficiency of FAPbBr₃ PeNCs, we applied the CMM incorporation strategy to FA_{0.9}GA_{0.1}PbBr₃ PeNCs where GA cation can induce much stronger hydrogen bonding with reduced surface defect. Even when GA is doped, the EL emission wavelength is 531 nm, which is very close to the green primary color (532 nm), showing the outstanding potential of our highly efficient CMM-PeNCs as next-generation pure green emitter (Fig. 5j).

3) A complete characterization of the perovskites employed is lacking. The authors only focus on the PL characterization in film, while they should present (at the very least):

Reply: Thank you for the valuable comment. We solidified our claim of lattice-strengthening more by strengthening PL characterization, EL characterization, and the investigations of the interaction between CMMs and PeNCs. Through supplementation of transient PL decay analysis with the PL spectrum varying the decay time, we confirmed more clearly that enhanced PL performance did not originate from the energy transfer between PeNCs and CMMs (Fig. 5d and Supplementary Fig. 23). Furthermore, we newly identified that near-unity PLQY could not be achieved simply by increasing interparticle spacing and reducing self-absorption (Supplementary Fig. 24). Additionally, we also analyzed the lattice-strengthening effect of TPBi in the solution state (Supplementary Fig. 25). Previous analysis of PLQY, transient PL, and temperature-dependent PL clearly showed the effect of lattice-strengthening on the luminescence (Fig. 2a-c, Fig. 5d, and Fig. 6a-b). Therefore, our PL characterization with newly added data demonstrated the importance of lattice-strengthening to improve the luminescent efficiency of perovskite.

As a complement to EL characterization, we additionally measured the operational lifetime of PeNC-CMM LEDs (Supplementary Fig. 31). TPBi CMM showed improvement in device-half lifetime due to the lattice-strengthening effect. Furthermore, the previous results showing improved EL efficiency by TPBi CMM are also evidence of lattice-strengthening (Fig. 5 and Supplementary Fig. 26). We showed that the EQE of PeNC-LED device was accurately

calculated based on the angle-dependent EL profile (Supplementary Fig. 28). Cross-sectional SEM image of LED device proved that the thickness used in our optical simulation was correct (Supplementary Fig. 27). Overall, the results of EL characterization as well as those of PL characterization obviously showed the effect of lattice-strengthening.

Furthermore, we comprehensively investigated various effects of CMMs such as crystal structure, particle size, film morphology, electronic structure, and interaction between CMMs and perovskite. The results of XRD measurement showed that the incorporation of CMMs did not affect the crystal structure of perovskite (Fig. 4a, and Supplementary Fig. 11,19). Through the TEM measurement, we confirmed that the average size and polydispersity of PeNCs were not significantly changed upon various CMM incorporation (Fig. 4b and Supplementary Fig. 12,21). The top-SEM images of perovskite indicated that the luminescent properties depending on the addition of CMMs had less relevance to the morphology of the films (Supplementary Fig. 13,14,20). Combined analysis of UPS spectra and UV-vis absorption spectra revealed that CMM did not significantly change the VBM and CBM of PeNCs (Fig. 4c and Supplementary Fig. 15,16). Through $^1\text{H-NMR}$ and FT-IR analysis, we experimentally proved the hydrogen bonding formation between CMMs and perovskite, which is a fundamental mechanism for lattice-strengthening (Fig. 4d,e).

In conclusion, we comprehensively investigated the impact of CMMs on perovskite including PL and EL characterizations. Those analyses evidently proved that luminescent efficiency is improved by lattice-strengthening. We also respond point-by-point to the comments regarding supplementation of characterization suggested by the reviewer, as shown below.

4) Optical absorption spectrum of both the ETL materials and the perovskites (bulk and NCs). This is important to understand if any form of energy transfer can take place between the two phases (e.g. TPBi absorbs in the blue spectral region). Similarly, optical absorption spectra and discussion of all the composite films is required.

Reply: We appreciate the reviewer's constructive comment related to energy transfer. We measured transient PL decay of $\text{FA}_{0.9}\text{GA}_{0.1}\text{PbBr}_3$ using 405 nm laser and 337 nm laser excitation (Fig. 5d and Supplementary Fig. 23). A laser power of 405 nm can only excite perovskites having a lower bandgap, but that of 337 nm laser can excite both CMMs and perovskites. Transient PL curves at both excitation conditions showed prolonged PL lifetime by TPBi and nearly similar PL decay trend. So, we concluded that enhanced PL performance did not originate from the energy transfer between CMM and PeNCs.

Furthermore, we additionally supplemented the PL spectrum of transient PL decay with varying the decay time using a streak camera (Supplementary Fig. 23b-d). If the energy transfer significantly occurs, it will be observed that the emission wavelength is red-shifted

with varying the decay time. However, our PeNCs with CMMs showed identical PL emission spectrum irrespective of the decay time, indicating that the energy transfer effect is negligible to the improvement of the luminescent performance.

Revision: Supplementary Fig. 16, 23 and its explanations

Supplementary Fig. 16:

Figure S16 | UV-vis absorption spectra. a, CMM films, b-d, Perovskite films with CMMs. bulk MAPbBr₃ (b), FAPbBr₃ PeNCs (c), FA_{0.9}GA_{0.1}PbBr₃ PeNCs (d).

Supplementary Fig. 23 and its explanation:

Figure S23 | Transient PL decay of $\text{FA}_{0.9}\text{GA}_{0.1}\text{PbBr}_3$ PeNC films with CMMs with 337 nm laser excitation. a, Transient PL decay, b-d, PL spectrum with varying the decay time. No CMM (b), PO-T2T (c), and TPBi (d).

Before: Transient PL decay with 337 nm N_2 laser excitation which excites CMMs also shows the same trend observed from the transient PL decay with 405 nm excitation shown in Figure 4d indicating that prolonged decay lifetime of TPBi treated PeNC didn't originate from the energy transfer between CMM and PeNCs.

After: Transient PL decay with 337 nm N_2 laser excitation which excites CMMs also shows the same trend observed from the transient PL decay with 405 nm excitation shown in Figure 5d indicating that prolonged decay lifetime of TPBi-treated PeNC didn't originate from the energy transfer between CMM and PeNCs. **In addition, an identical PL spectrum irrespective of the decay time indicated that the energy transfer had little effect on the enhanced luminescence mechanism.**

5) Characterization of the perovskite NCs in solution. Do the ETL materials impact the PL/ABS in solution as well? If they truly induce lattice-strengthening, such effect should be visible in solution too (maybe employing a higher concentration of ETLs)

Reply: Thank you very much for the instructive comments. After preparing PeNC and TPBi solution separately, we blended the two solutions to simplify the lattice-strengthening process (Fig.1). In order to characterize lattice-strengthening effect in the solution phase, we precisely controlled the concentration of pure TPBi solution and measured PL/ABS of TPBi-blended PeNC solutions (Supplementary Fig. 25). With increasing TPBi content in the PeNC/TPBi mixture solutions, the PLQY of that was slightly improved. However, the effect was not significant because molecules and ligands in the solution phase move relatively freely and

continuously detach and attach (*ACS Nano* **10**, 2071-2081 (2016)). Therefore, the extent of improving PLQY was lower than that of the film (Supplementary Fig. 24). Nevertheless, when spin-coating $\text{FA}_{0.9}\text{GA}_{0.1}\text{PbBr}_3$ PeNC solution containing TPBi, TPBi CMM anchors at the perovskite surface, where the organic ligands are not passivating, enabling efficient lattice-strengthening.

Revision: Supplementary Fig. 25, its explanation and Reference

Figure S25 | Effect of TPBi CMM on colloidal $\text{FA}_{0.9}\text{GA}_{0.1}\text{PbBr}_3$ PeNC solution depending on the concentration of TPBi solution. a, PLQY. b, UV-vis absorption spectra. c, PL spectrum.

To confirm the presence of lattice-strengthening effect in the colloidal PeNC solution, we measured UV-vis absorption spectra and PLQY of $\text{FA}_{0.9}\text{Pb}_{0.1}\text{Br}_3$ -TPBi mixed solutions of different TPBi concentrations. Incorporating TPBi CMM yielded a slight increase in the PLQY of colloidal $\text{FA}_{0.9}\text{GA}_{0.1}\text{PbBr}_3$ PeNC solution, but the effect was not significant because molecules and ligands in the solution phase move relatively freely and continuously detach and attach⁵. Therefore, the extent of the lattice-strengthening effect was lower in the solution than in the film. However, when spin-coating colloidal $\text{FA}_{0.9}\text{GA}_{0.1}\text{PbBr}_3$ solution containing TPBi CMM, the TPBi CMM anchors at the perovskite surface where the organic ligands are not passivating, leading to more effective lattice strengthening (Supplementary Fig. 24).

Reference

5. De Roo, J. *et al.* Highly Dynamic Ligand Binding and Light Absorption Coefficient of Cesium Lead Bromide Perovskite Nanocrystals. *ACS Nano* **10**, 2071–2081 (2016).

6) How were the non-radiative rates estimated? Looking at the TRPL, these are multi-exponential decays (how were they fit?), the authors need to explain how they extract such numbers. Extraction of these values is not as trivial as the authors make it seem, in particular if some resemblance of physical meaning is to be retained.

Reply: Thank you very much for the constructive comments. Our TrPL results indeed showed multi-exponential decay. Previously, we defined average lifetime (τ_{PL}) as the time to reach 99% emission of majority carriers and calculated non-radiative decay rates. However, in the field of perovskite optoelectronics whose emitters show multi-exponential PL decays, the tri-exponential fitting method was generally used to analyze transient PL decay characteristics

with physical meanings of monomolecular recombination rate (k_1), bimolecular recombination rate (k_2), and auger recombination rate (k_3) (*Acc. Chem. Res.* **49**, 146-154 (2016)).

$$\frac{dn}{dt} = -k_1n - k_2n^2 - k_3n^3$$

Therefore, we re-calculated the average lifetime of bulk perovskite and PeNCs from the transient PL decay curves using tri-exponential decay functions (Fig. 2b,c, Fig. 5d, Supplementary Fig. 7). Through accurately measured PLQY and the average lifetime, we could obtain radiative decay rate (k_r) and non-radiative decay rate (k_{nr}) of FAPbBr₃ PeNCs based on the relations below (*J. Am. Chem. Soc.* **140**, 2656-2664 (2018)) (Fig. 2a and Supplementary Table. 1).

$$\text{PLQY} = k_r / (k_r + k_{nr})$$

$$k_r + k_{nr} = 1 / \tau_{PL}$$

We also compared the previous and re-calculated values of k_r and k_{nr} . As can be seen below, in both calculated values, the decreasing trend of non-radiative decay rates according to the addition of CMM was the same.

Condition	PLQY (%)	Previous values			Re-calculated values		
		τ_{avg} (ns)	k_r (10^6 s^{-1})	k_{nr} (10^6 s^{-1})	τ_{avg} (ns)	k_r (10^6 s^{-1})	k_{nr} (10^6 s^{-1})
no CMM	79.5	79	10.06	2.59	20.5	38.7805	10
3TPYMB	83.9	83	10.11	1.94	21.6	38.8426	7.45370
PO-T2T	87.2	87	10.02	1.47	24.2	36.0331	5.28926
TPBi	95.2	100	9.52	0.48	27.4	34.7445	1.75182

Furthermore, in order to ensure that readers understand this calculation process, we added a description for calculating average lifetime and nonradiative decay rates. Additionally, we supplemented appropriate references in the method section.

Revision: Page 5 line 18 in the revised manuscript, Supplementary Fig. 7, Supplementary Table. 1, Method and Reference

Page 5 line 18 in the revised manuscript:

Before: The average PL decay lifetime became longer and nonradiative decay rates were decreased, indicating that CMMs suppressed exciton quenching, both for PeNCs and for nanograined bulk perovskite films (Fig. 2a-c). For example, the average PL decay lifetime of FAPbBr₃ PeNC films increased from 79 ns without CMM to 83 ns, 87 ns, and 100 ns for 3TPYMB, PO-T2T, and TPBi incorporated films. Accordingly, the nonradiative decay rates of FAPbBr₃ PeNC films reduced from

$2.59 \times 10^6 \text{ s}^{-1}$ to $1.92 \times 10^6 \text{ s}^{-1}$, $1.47 \times 10^6 \text{ s}^{-1}$, and $0.50 \times 10^6 \text{ s}^{-1}$.

After: The average PL decay lifetime became longer and nonradiative decay rates were decreased, indicating that CMMs suppressed exciton quenching, both for PeNCs and for nanograined bulk perovskite films (Fig. 2a-c, **Supplementary Fig. 7** and **Supplementary Table. 1**). For example, the average PL decay lifetime of FAPbBr₃ PeNC films increased from 20.5 ns without CMM, to 21.6 ns in film with 3TPYMB, 24.2 ns in film with PO-T2T, and 27.4 ns in film with TPBi incorporated films. Accordingly, the nonradiative decay rates of FAPbBr₃ PeNC films decreased from $10.0 \times 10^6 \text{ s}^{-1}$ with no CMM to $7.45 \times 10^6 \text{ s}^{-1}$ with 3TPYMB, $5.29 \times 10^6 \text{ s}^{-1}$ with PO-T2T, and $1.75 \times 10^6 \text{ s}^{-1}$ with TPBi.

Supplementary Fig. 7 and Supplementary Table. 1:

Figure S7 | Transient PL decay of FAPbBr₃ PeNC films with tri-exponential fittings. a, no CMM, b, 3TPYMB, c, PO-T2T, and d, TPBi.

Table S1 | Photophysical properties of FAPbBr₃ PeNCs with CMMs. Average PL decay lifetime (τ_{PL}) was obtained from transient PL decay curves by using tri-exponential fitting. Radiative decay rates (k_r) and nonradiative decay rates (k_{nr}) were calculated using the values of PLQY and τ_{PL} based on the following relations. $PLQY = k_r / (k_r + k_{nr})$. $k_r + k_{nr} = 1/\tau_{PL}$.

Condition	A ₁	τ_1 (ns)	A ₂	τ_2 (ns)	A ₃	τ_3 (ns)	τ_{PL} (ns)	PLQY (%)	k_r (10^6 s^{-1})	k_{nr} (10^6 s^{-1})
no CMM	0.37593	1.52461	0.40463	13.68738	0.16871	79.07663	20.5	79.5	38.7805	10
3TPYMB	0.3899	1.90767	0.40711	15.52381	0.16344	83.87618	21.6	83.9	38.8426	7.45370
PO-T2T	0.35165	2.00532	0.41589	16.67015	0.18268	83.81514	24.2	87.2	36.0331	5.28926
TPBi	0.38975	2.84804	0.41396	21.04269	0.16515	101.2882	27.4	95.2	34.7445	1.75182

Method and Reference:

Before:

Photoluminescence and photoluminescence quantum efficiency measurement

PLQY of the PeNC films were measured with an integrating sphere and PMT detector using 325 nm CW He:Cd laser as the excitation source. The integrating sphere was purged with nitrogen gas during the measurement of PLQY. The PL spectra of encapsulated PeNC films were measured with a CCD spectrometer (Maya2000, Ocean Optics) using 325 nm CW He:Cd laser as the excitation source. PL spectra and PLQY of bulk perovskite films were measured with a JASCO FP8500 spectrofluorometer with a xenon arc lamp with 405 nm as an excitation source. Transient PL decay of perovskite films were obtained by a TCSPC set-up (FluoTime 300 (PicoQuant)) composed of a 405 nm pulsed laser as an excitation source (LDH-P-C-405B, PicoQuant), photon-counting detector (PMA Hybrid 07) and a TCSPC module (PicoHarp, PicoQuant).

After:

Analysis for photoluminescence characteristics

PLQY of the PeNC films were measured with an integrating sphere and PMT detector using 325 nm CW He:Cd laser as the excitation source. The integrating sphere was purged with nitrogen gas during the measurement of PLQY. The PL spectra of encapsulated PeNC films were measured with a CCD spectrometer (Maya2000, Ocean Optics) using 325 nm CW He:Cd laser as the excitation source. PL spectra and PLQY of bulk perovskite films were measured with a JASCO FP8500 spectrofluorometer with a xenon arc lamp with 405 nm as an excitation source. Transient PL decay of perovskite films were obtained by a TCSPC set-up (FluoTime 300 (PicoQuant)) composed of a 405 nm pulsed laser as an excitation source (LDH-P-C-405B, PicoQuant), a photon-counting detector (PMA Hybrid 07) and a TCSPC module (PicoHarp, PicoQuant). **The average PL decay lifetimes of bulk perovskite and PeNC films were obtained by fitting transient PL curves using tri-exponential decay functions⁴⁷. The nonradiative decay rate (k_{nr}) was calculated based on the relation below⁴⁸.**

$$PLQY = k_r / (k_r + k_{nr})$$

$$k_r + k_{nr} = 1 / \tau_{PL}$$

Reference

47. Johnston, M. B. & Herz, L. M. Hybrid Perovskites for Photovoltaics: Charge-Carrier Recombination, Diffusion, and Radiative Efficiencies. *Acc. Chem. Res.* **49**, 146–154 (2016).
48. Imran, M. *et al.* Benzoyl Halides as Alternative Precursors for the Colloidal Synthesis of Lead-Based Halide Perovskite Nanocrystals. *J. Am. Chem. Soc.* **140**, 2656–2664 (2018).

7) Do the ETL materials affect the stability of the perovskites? Stability is completely neglected in the manuscript while this is currently one of the most important figure-of-merit for perovskites. Moreover, improved stability is a real fingerprint of increased lattice-strengthening as the soft nature of perovskites is one of the main reasons of their low functional lifetime.

Reply: We appreciate the reviewer bringing up this important point. We measured the operational lifetime of colloidal FA_{0.9}GA_{0.1}PbBr₃ PeNC-LEDs with no CMM, TPBi, and PO-T2T at an initial luminance (L_0) of 100 cd/m², as shown in Supplementary Fig. 31. The device with

no CMM has device half-lifetime (LT_{50}) of 1.6 h, but TPBi CMM exhibited a 3.1-fold enhancement of the operational lifetime (LT_{50} of 5 h). TPBi CMM also improved luminescent efficiency by lattice-strengthening without ligand detachment effect (Fig. 3,5). On the other hand, the device with PO-T2T CMM showed decreased stability (LT_{50} of 0.7 h). This result can be attributed to the ligand detachment effect studied in our manuscript, as PO-T2T has a strong interaction with oleic acid (Fig. 6). The results demonstrate that our lattice-strengthening strategy by TPBi CMM that does not induce ligand detachment not only improves efficiency but also enhances stability.

Revision: Page 10 line 26 in the revised manuscript, Supplementary Fig. 31, and its explanation

Page 10 line 26 in the revised manuscript:

In addition to the improvement of efficiency, the TPBi CMM device showed a longer device half-lifetime by a factor of 3.1 compared to the device without CMM, indicating that lattice-strengthening by TPBi that does not induce ligand detachment also contributes to suppressing the degradation process of the perovskite (Supplementary Fig. 31).

Supplementary Fig. 31 and its explanation:

Figure S31 | Operational lifetime of colloidal FA_{0.9}GA_{0.1}PbBr₃ PeNC devices with CMMs.

To investigate the effect of CMM on the stability of colloidal PeNC-LEDs, we measured the device half-lifetime (LT_{50}) of colloidal FA_{0.9}GA_{0.1}PbBr₃ PeNC-LEDs under the condition that the initial luminance was 100 cd/m². Compared to no CMM device (LT_{50} of 1.6h), the TPBi CMM device showed improved device lifetime (LT_{50} of 5h). On the other hand, PO-T2T exhibited reduced device stability (LT_{50} of 0.7h) because it has highly reactive characteristics with oleic acid ligand (Fig. 6). Therefore, we demonstrated that lattice-strengthening by TPBi CMM that does not induce ligand detachment not only enhances luminescent efficiency, but suppresses the degradation of perovskite.

8) Please include TEM images of the NCs as well, similar to the SEM shown for the MAPbBr film. The authors should also rule out that the improve in PLQY in the NC film is not due to increased spacing between the NCs caused by the ETL additives (i.e., are these NCs mono-

disperse? Do poly dispersity change after the ETL addition? Does self-absorption change in the film upon the ETL addition?). Some PL spectra report a small blue-shift (see figure S12) which could indicate a change in self-absorption.

Reply: Thank you very much for inquiring these questions. We additionally measured TEM images of colloidal FAPbBr₃ PeNCs and FA_{0.9}GA_{0.1}PbBr₃ PeNCs with CMMs (Supplementary Figs. S12 and S21a-c). The colloidal PeNCs are extremely beam-sensitive to the electron beam in TEM, showing significant changes in morphologies at high accumulated doses (*Science* **359**, 675-679 (2018)). Therefore, in order to obtain good images as much as we can, we used a TEM instrument equipped with a cold field emission gun, an image corrector, and a Gatan K3 IS direct electron detector. Moreover, we also delicately found the regions of interest and adjusted image focus as quickly as possible with a small electron dose rate of $\sim 1 e^-/\text{Å}^2\cdot\text{s}$, and acquired TEM images with an electron dose of $10 e^-/\text{Å}^2\cdot\text{s}$ using K3 IS detector at the optimized condition with negligible structural changes of colloidal PeNCs while securing the signal-to-noise ratio of the images.

We obtained particle size distribution from TEM images of each condition and calculated the average particle size (Fig. 4b and Supplementary Fig. 21d). Colloidal FAPbBr₃ PeNCs with no CMM, 3TPYMB, PO-T2T, and TPBi showed the average particle size of 9.78 nm, 9.90 nm, 9.98 nm, and 9.88 nm, respectively. Also, colloidal FA_{0.9}GA_{0.1}PbBr₃ PeNCs with no CMM, PO-T2T, and TPBi showed the average particle size of 9.94 nm, 9.75 nm, and 9.94 nm, respectively. Both colloidal FAPbBr₃ and FA_{0.9}GA_{0.1}PbBr₃ PeNCs showed negligible change in their average particle size while maintaining a narrow distribution (poly-dispersity). Therefore, we conclude that CMM incorporation does not much affect the particle size of colloidal PeNCs.

In order to analyze the effect of interparticle spacing and self-absorption, we employed triphenylbenzene (TPB) consisting of only conjugated benzene rings without a functional group (Supplementary Fig. 24). When the concentration of TPB solution increased, the interparticle spacing could be extended, but this molecule hardly induces lattice-strengthening because it has no functional groups (e.g. benzimidazole group). We compared the PLQY of FA_{0.9}GA_{0.1}PbBr₃ PeNC films depending on the concentration of TPBi and TPB solutions. We confirmed that increasing the concentration of TPB solution improved PLQY up to 86.3% at 10mM, but the extent of enhancement was lower than TPBi and improvement is nearly saturated. Moreover, in the PL spectrum of FA_{0.9}GA_{0.1}PbBr₃ with TPB (Supplementary Fig. 24c), a gradual blue-shifted PL peak (~ 1 nm shift) was observed, indicating that increased TPB concentration enabled self-absorption to be lower. Overall, these results reflected that PLQY of PeNC films can be improved by increasing inter-particle spacing, but only decreasing self-absorption could not achieve near-unity PLQY. Since we proved that particle size

distribution is nearly the same (Supplementary Fig. 21), the blue-shifted PL peak by embedding TPBi CMM (~ 1 nm shift) (Supplementary Fig. 22) could be attributed to defect passivation or reduced self-absorption. In addition, the red-shifted PL peak in PO-T2T might be due to the formation of metallic Pb defects (Fig. 6c).

Revision: Page 8 line 19 in the revised manuscript, Fig. 4b, Supplementary Fig. 12, Page 9 line 29 in the revised manuscript, Supplementary Fig. 21, Supplementary Fig. 24 and its explanations.

Page 8 line 19 in the revised manuscript, Fig. 4b, and Supplementary Fig. 12

High-resolution transmission electron microscopy (TEM) images of colloidal FAPbBr_3 PeNCs showed average particle size of ~9.78 nm without CMM, 9.90 nm with 3TPYMB, 9.98 nm with PO-T2T, and 9.88 nm with TPBi (Fig. 4b and Supplementary Fig. 12). This result indicates that post-synthesis blending of CMMs into colloidal PeNCs has negligible effect on the average particle size and distribution.

Fig. 4 | Interaction between FAPbBr_3 PeNC and CMMs. a-c, Independent characteristics of FAPbBr_3 PeNCs with CMMs. The size distribution (**b**) of colloidal FAPbBr_3 PeNCs. Particle size was extracted from transmission electron microscope (TEM) images.

Figure S12 | TEM images of colloidal FAPbBr₃ PeNCs with CMMs. a, no CMM, b, 3TPYMB, c, PO-T2T and d, TPBi. Each image size is 40 nm × 40 nm.

Page 9 line 29 in the revised manuscript and Supplementary Fig. 21:

Similar to the studies of FAPbBr_3 PeNCs (Fig. 4a,b and Supplementary Figs. 12,13), CMM incorporation had negligible impact on the crystal structure, film morphology, and particle size of GA-doped FAPbBr_3 PeNCs (Supplementary Figs.19-21).

Figure S21 | TEM images of colloidal $\text{FA}_{0.9}\text{GA}_{0.1}\text{PbBr}_3$ PeNCs with CMMs. a, no CMM, b, PO-T2T, c, TPBi, and d, size distributions. Each image size is $40 \text{ nm} \times 40 \text{ nm}$.

Supplementary Fig. 24 and its explanations:

Figure S24 | PLQY of FA_{0.9}GA_{0.1}PbBr₃ PeNC films depending on the concentration of CMM solutions. a, TPBi CMM. b-c, Triphenylbenzene (TPB). PLQY (b) and PL spectrum (c) of FA_{0.9}GA_{0.1}PbBr₃ with TPB.

To investigate how the content of CMMs influences PLQY, we adjusted the concentration of CMM solutions. As the concentration of TPBi solution increased, the effect of lattice-strengthening increased. The PLQY was highest (near-unity) when TPBi solution was 5 mM. To clarify that the improved PLQY is due to lattice-strengthening rather than reduced self-absorption related to inter-particle spacing, triphenylbenzene (TPB), which has no functional group, was incorporated into FA_{0.9}GA_{0.1}PbBr₃ PeNC. As the concentration of TPB was increased, the PLQY of FA_{0.9}GA_{0.1}PbBr₃ increased slightly, and concurrently the PL spectrum was blue-shifted (by ~1 nm), possibly because self-absorption was reduced by increased inter-particle spacing. However, near-unity PLQY could not be achieved by only the reduced self-absorption effect. These contrasting results of TPBi and TPB highlight the importance of lattice-strengthening at the perovskite surface to increase luminescent efficiency.

9) Regarding the LEDs, many important characterizations are missing in my opinion. First of all, can the authors describe in the experimental method how the EQE was measured? The reported EQE is obtained through an integrating sphere? Or it is referring to only the light emitted in the forward direction?

Reply: We appreciate the reviewer for presenting the comment to solidify our experimental result. We supplemented the angular electroluminescence intensity profile in Supplementary Fig. 28. The emission profile of our PeLED is closely matched with the Lambertian profile, and EQE was accurately calculated based on the resulted angle-dependent EL profile (*Joule* **4**, 1206-1235 (2020)). We also added the information related to EQE calculation in the section of Methods.

Revision: Page 10 line 13 in the revised manuscript, Reference, Supplementary Fig. 28, and Methods

Page 10 line 13 in the revised manuscript:

Before: The CMM treatment on GA-doped PeNCs results in more efficient PeNC-LED with an EQE of 26.1%, which is the highest efficiency among published reports of PeLEDs using colloidal PeNCs without outcoupling enhancement technique (Fig. 4j and Supplementary Table. 2).

After: The CMM treatment on GA-doped PeNCs results in a more efficient colloidal PeNC-LED with

an EQE of 26.1%, calculated based on angular electroluminescence distribution⁴⁵ (Supplementary Fig. 28). Our colloidal PeNC-LEDs based on this materials design approach achieved the highest efficiency among published reports of PeLEDs using colloidal PeNCs without an outcoupling enhancement technique (Fig. 5j, Supplementary Fig. 29 and Supplementary Table. 2,3).

Reference

45. Jeong, S. H. *et al.* Characterizing the Efficiency of Perovskite Solar Cells and Light-Emitting Diodes. *Joule* **4**, 1206–1235 (2020).

Supplementary Fig. 28:

Figure S28 | Angle-dependent EL profile for colloidal FA_{0.9}GA_{0.1}PbBr₃ PeNC-LED with TPBi.

Method:

Electroluminescence characterization

The current density-voltage-luminescence (J - V - L) characteristics of PeLEDs were measured using a spectroradiometer (CS-2000, Minolta) and an electrical source-measurement unit (Keithley 236). The EQE of PeLEDs was calculated based on the angle-dependent EL profile⁴⁵. Transient EL analysis was performed using a streak camera system composed of a streak scope (C10627, Hamamatsu Photonics), a CCD camera (C9300, Hamamatsu Photonics), a delay generator (DG645, Stanford Research Systems), and a function generator (33220A, Agilent).

10) The authors completely neglect that the ETL addition can change the energy levels of the active layer. The authors should perform a detailed UPS study of the films for the various compositions employed to understand how the additives are changing the VBM/CBM of the system. This is to verify how the energy levels are changing and how this could affect charge transport in the device.

Reply: Thank you very much for the beneficial comments. We will respond the comments #10,11 as one part since they are closely related.

11) In addition to the UPS, the authors must discuss how the electrical properties of the films

with ETL additives are different from the pristine one. Here, we are talking of adding relatively conductive molecules, I would expect some considerable changes in the charge transport properties. Do the films become more n-type? Does the carrier density increase? How can the authors assign the observed effects to lattice-strengthening while they completely disregard the electrical properties? The authors should produce electron and hole only devices and study how the charge transport change.

Reply: Thank you very much for the beneficial comments. We measured UPS spectra of FAPbBr₃ PeNC films with CMMs (Supplementary Fig. 15). The VBM of PeNC-CMM was calculated from the secondary cut-off and onset values of UPS spectra. Furthermore, we additionally measured the UV-vis spectrum to obtain the optical band gap of FAPbBr₃ PeNCs (Supplementary Fig.16). Based on both results, we calculated VBM/CBM levels of FAPbBr₃ PeNCs with embedding CMMs and illustrated electronic band structures of them in Fig. 4c. VBM levels of FAPbBr₃ PeNCs with incorporating CMMs were the same, but only PO-T2T showed work function value as low as 0.17 eV due to increased secondary cut-off. This means FAPbBr₃ PeNC with PO-T2T CMM has slightly more n-type characteristics than others. Reduced work function can be attributed to higher electronegativity of O atoms in PO-T2T (*Nature* **611**, 688-694 (2022)) or surface modification of PeNCs by PO-T2T having strong interaction with oleic acid ligand (Fig. 6) (*J. Am. Chem. Soc.* **140**, 10504–10513 (2018), *Nature Electronics* **3**, 704-710 (2020)). However, slight n-type doping by PO-T2T CMM did not increase the current density of the PeNC-LED device. Rather, the incorporation of CMMs (3TPYMB, PO-T2T, and TPBi) reduced the current density of PeNC-LEDs (Supplementary Fig. 8d).

To figure out the reason for the decrease in the current density by CMMs, we fabricated hole-only and electron-only devices, and compared their current density (Supplementary Fig. 17). While the current density of electron-only device with embedding CMMs was almost the same, CMM incorporation significantly reduced the current density of hole-only devices. Moreover, the decreasing trend of the current density in hole-only devices coincided well with that of actual PeNC-LED devices. We also found that the current density of the hole-only device decreased as the HOMO level of CMMs was deepened. Therefore, the reduced current density of PeNC LEDs originated from decreased hole-current density due to the hole-blocking characteristics of electron-transporting CMMs. Nevertheless, we are confident that lattice-strengthening is the dominant mechanism to improve luminescent efficiency based on the following reasons. (1) FAPbBr₃ PeNC films showed different PL performance, which is consistent with improved elastic modulus by CMMs. (2) PL and EL performance are well-matched with each other.

Revision: Page 8 line 25 in the revised manuscript, Fig. 4c, Supplementary Fig. 15-17, and their explanations.

Page 8 line 25 in the revised manuscript and Fig. 4c:

Combined analysis of ultraviolet photoelectron spectroscopy (UPS) spectra and UV-vis absorption spectra confirmed that all CMMs do not affect the valence band maximum (VBM) or conduction band maximum (CBM) of PeNCs (Fig. 4c and Supplementary Figs. 15,16). The work functions were not changed by CMMs of TPBi and 3TPYMB, but the incorporation of PO-T2T CMM lowered the work function by 0.17 eV, making FAPbBr₃ PeNC with PO-T2T slightly more n-type than the FAPbBr₃ PeNC without CMM. However, the effect of CMMs on the current density of the electron-only device was negligible despite the electron-transporting nature of the CMMs (Supplementary Fig. 17). On the other hand, due to the hole-blocking characteristics of the CMMs, the current density of the hole-only device was largely decreased (Supplementary Fig. 17). This led to the decreased current density of colloidal FAPbBr₃ PeNC-LEDs with PO-T2T and TPBi CMMs compared to that of the device without CMM. Thus, as the CMMs do not significantly impact the crystal structure, particle size distribution, film morphology, VBM, and CBM of PeNCs, the lattice strengthening effect is the primary contributor to the improved luminescent efficiency.

Fig. 4 | Interaction between FAPbBr₃ PeNC and CMMs. a-c, Independent characteristics of FAPbBr₃ PeNCs with CMMs. Electronic structure (c) of FAPbBr₃ PeNCs. Energy band diagrams of FAPbBr₃ PeNCs with CMMs were obtained by combining ultraviolet photoelectron spectroscopy (UPS) analysis and optical bandgap. Valence band maximum (VBM) and optical bandgap were derived from UPS spectra and UV-vis absorption spectra, respectively. The conduction band maximum (CBM) was calculated based on the VBM level and optical bandgap of PeNCs.

Supplementary Figs. 15,16:

Figure S15 | UPS spectra of FAPbBr₃ PeNC films with CMMs.

Figure S16 | UV-vis absorption spectra. **a**, CMM films, **b-d**, Perovskite films with CMMs. bulk MAPbBr₃ (**b**), FAPbBr₃ PeNCs (**c**), FA_{0.9}GA_{0.1}PbBr₃ PeNCs (**d**).

Supplementary Fig. 17 and its explanation:

Figure S17 | Hole-only and electron-only device based on colloidal FAPbBr₃ PeNCs with CMMs. a, hole-only device, b, electron-only device, c, energy band diagram of FAPbBr₃ PeNCs, TPBi, and PO-T2T.

To investigate how electron-transporting CMMs affect current density in colloidal PeNC-LEDs, we compared the current density of hole-only and electron-only devices. The structure of hole-only devices was ITO (70 nm)/GraHIL (60 nm)/FAPbBr₃ PeNC (30 nm)/ 4,4',4''-tris(*N*-carbazolyl)-triphenylamine (TCTA) (50 nm)/Molybdenum(VI) oxide (MoO₃) (5 nm)/Al (100 nm). The structure of electron-only device was ITO (70 nm)/polyethyleneimine ethoxylated (PEIE) (10 nm)/FAPbBr₃ PeNC (30 nm)/TPBi (45 nm)/LiF (1 nm)/Al (100 nm). Although PO-T2T lowered the work function of FAPbBr₃ PeNCs by 0.17 eV (Fig. 3c and Supplementary Fig. 13), electron-only devices with no CMM, with TPBi and with PO-T2T showed similar current density. On the contrary, adding TPBi and PO-T2T into FAPbBr₃ PeNCs reduced the current density of hole-only devices. Moreover, as the HOMO level of CMMs was deepened, the current density of hole-only devices decreased more. Therefore, reduced current density in FAPbBr₃ PeNC-LEDs can be attributed to decreased hole current by the hole-blocking characteristics of CMMs (Supplementary Fig. 8d).

12) An energy band diagram and a device cross-section should be provided.

Reply: Thank you very much for the comment to solidify our study. Based on the UPS study (Fig. 4c and Supplementary Fig. 15), we revised the energy band diagram showing with device structure (Supplementary Fig. 3). Furthermore, we measured cross-section SEM image of our PeLED (Supplementary Fig. 27), and the thickness value of each layer matched with the thickness we described in Supplementary Fig. 3.

Revision: Supplementary Fig. 3, 27 and the explanation

Supplementary Fig. 3:

Figure S3 | Device structure of PeLEDs. a, Bulk perovskite, b, PeNC

Supplementary Fig. 27 and its explanation:

Figure S27 | Cross-sectional SEM image of colloidal FA_{0.9}GA_{0.1}PbBr₃ PeNC-LED.

To ensure that the thickness of each layer in the device was correct, we measured a cross-SEM image of colloidal FA_{0.9}GA_{0.1}PbBr₃ PeNC-LED. We confirmed that the thickness of FA_{0.9}GA_{0.1}PbBr₃ PeNC is 30 nm, which is consistent with the thickness measured by ellipsometry (Supplementary Fig. 4). In addition, the thickness of the other layers was found to be consistent with the thickness we described (Supplementary Fig. 3).

13) Once again, stability is completely disregarded, the authors should provide data for the functional lifetime of their devices and compare it against a pristine reference. Many examples of these measurements are reported in the literature.

Reply: We appreciate the reviewer bringing up this important point again. We completely agree with the reviewer's comment that it is important to investigate the functional lifetime of

the device. However, we noticed that this comment is similar to comment #7, and we responded to this comment in detail in the section of comment-response #7. Therefore, we would appreciate it if the reviewer referred to comment-response #7.

14) The authors claim to have a nearly perfect device limited only by optical outcoupling. Convincing evidence must be provided, for example the authors must demonstrate that they have nearly reached perfect charge balancing in the active layer.

Reply: Thank you for the detailed comment. The theoretically achievable EQE of an LED is the [outcoupling efficiency] × [charge balance factor]. The outcoupling efficiency can be calculated by optical simulation using experimentally obtained input parameters including PLQY of the emitting layer, PL spectrum of the emitting layer, device structure (thickness), and refractive index of layers. However, the charge balance factor of an LED device cannot be directly measured. Hence, the optically achievable EQE of an LED is calculated by optical simulation by assuming a perfect charge balance. We compared EQE of our device (26.1%) with theoretically achievable EQE (29.4%) from optical simulation (Fig. 5i) to show that our lattice strengthening approach leading to near-unity PLQY was also effectively applied to the device fabrication. In order to prevent misunderstanding, we also changed the tone of the sentence related to optical simulation in the revised manuscript.

Reference

- *Journal of Photonics for Energy* **1**, 011006 (2011)

- *Phys. Status Solidi A* **210**, 44-65 (2013)

Revision: Page 10 line 24 in the revised manuscript

Before: Moreover, the fabricated device showed a high EQE approaching the theoretical efficiency obtained from the optical simulation assuming a perfect charge balance and a PLQY of unity (Fig. 4i and Supplementary Fig. 15).

After: Moreover, the fabricated device showed a high EQE (26.1 %) close to the theoretical efficiency (29.4%) obtained from optical simulation assuming a perfect charge balance and a PLQY of unity (Fig. 5i and Supplementary Fig. 30).

15) The manuscript is quite interesting, but it lacks a considerable amount of experimental data to support the lattice-strengthening claimed by the authors. The modelling is not sufficient to sustain their claims in my view.

Reply: We appreciate the reviewer for the constructive comments to improve our manuscript again. Through this revision, we supplemented much of the evidence that can support our

claim regarding lattice-strengthening. Through $^1\text{H-NMR}$ and FT-IR analysis, we experimentally proved the hydrogen bonding formation between CMMs and perovskite, which is the fundamental mechanism for lattice-strengthening (Fig. 4d,e). In addition to the new experimental results above, experimentally measured Raman spectra clearly showed lattice-strengthening of perovskite (Fig. 3h). The calculated Raman spectra matched well with experimentally measured Raman spectra, indicating that our theoretical calculations were also designed very accurately (Fig. 3g and Supplementary Fig. 10). We also calculated elastic modulus of lattice-strengthened perovskite by CMMs, which clearly explained the results about luminescent performance of perovskite films and devices (Fig. 2,3,5). Improved operation lifetime of PeNC-LED by TPBi CMM is also one of the evidences for lattice-strengthening (Supplementary Fig. 31). We also demonstrated that near-unity PLQY through lattice-strengthening by TPBi CMM is not due to changes in self-absorption or energy transfer (Supplementary Fig. 23,24).

Moreover, we comprehensively investigated various effects of CMMs on the perovskite such as crystal structure, particle size, film morphology, electronic structure, and interaction. The results of XRD measurement showed that the incorporation of CMMs does not affect crystal structure (Fig. 4a, and Supplementary Fig. 11,19). Through the TEM measurement, we confirmed that the average size and polydispersity of PeNCs were not significantly changed upon CMM incorporation (Fig. 4b and Supplementary Fig. 12,21). The top-SEM images of perovskite indicated that the luminescent properties according to the addition of CMMs have less relevance to the morphology of the films (Supplementary Fig. 13,14,20). Combined analysis of UPS spectra and UV-vis absorption spectra revealed that CMM does not change the CBM and VBM of PeNCs (Fig. 4c and Supplementary Fig. 15,16). We clearly proved that those effects from CMM incorporation are not the primary mechanism of improving luminescence efficiency in our system.

In conclusion, we obviously identified the formation of hydrogen bonding, which is the fundamental mechanism for lattice-strengthening. Through Raman analysis, we observed that the vibrational peak related to the perovskite lattice shifted to a higher frequency, which is crucial evidence of lattice-strengthening (Fig. 3g,h). The calculated elastic modulus of perovskite with CMMs precisely matched with the trends of luminescent performance depending on CMMs (Fig. 2 and Fig. 3a-f). We thoroughly investigated various effects that can be induced by CMM incorporation (Fig. 4), solidifying that luminescent efficiency is improved by lattice-strengthening rather than other effects. Therefore, we are confident that CMM strengthens perovskite lattice, leading to improvement of luminescent performance.

Reviewer #3 (Remarks to the Author):

In this paper, Kim et al. reports that they improve the EQE of LED device based on perovskite nanocrystals by using conjugated molecular multipods (CMMs) to passivate the nanocrystal surfaces. Although many papers about surface passivation to improve LED efficiency have been reported, surface passivation related to the dynamic lattice disorder in this paper is an original concept. However, there are still some problems should be solved before this paper is published in Nature Communications.

1) The authors claim that hole transporting molecules would result in exciplex formation in EL devices (Supplementary Fig. 1-3). How to rule out the parasitic emission from the perovskite layer and not the transport layer? In addition, this discussion is based on the bulk perovskite materials, how about perovskite nanocrystals? In perovskite nanocrystals case, is there direct evidence for parasitic emission in EL devices when conjugated hole transporting molecules passivate nanocrystal surface?

Reply: Thank you very much for the valuable comment. First, we used hole-transporting molecules in both cases of bulk perovskite (Supplementary Fig. 1a,b) and perovskite nanocrystal (Supplementary Fig. 1c). When we fabricated PeLEDs with hole-transporting CMMs in both systems (bulk perovskite and PeNC) (Supplementary Fig. 1), the exciplex and electromer emission were observed in the EL devices. We used the same device structure of PeLEDs with electron-transporting CMMs (Fig. 2d-f, Supplementary Fig. 3,8). As can be seen in Fig. 2d, there is no parasitic emission in our devices with electron-transporting CMMs, meaning that the structure of the device is carefully designed so that the recombination zone is located precisely in the active emitting layer (Perovskite with CMMs), not electron-transporting layer (ETL). Referring to the literature, EL emission of TPBi showed the peak around 400 nm (*APL Mater.* **4**, 046102 (2016), *Organic Electronics* **17**, 377-385 (2015)). Therefore, the EL emission around 475 nm does not come from the TPBi ETL (Supplementary Fig. 1).

The exciplex emission around 475 nm in the EL device with CMMs of TAPC and TPD showed a similar spectrum shape irrespective of the hole transporting CMMs (Supplementary Fig. 1a,b). To clarify the EL emission around 475 nm, we fabricated films of TAPC, TPD, TPBi, and blends of them (Supplementary Fig. 2). The shape of EL emission around 475 nm closely matched with that of the exciplex emission from TAPC/TPBi and TPD/TPBi blend films. Additionally, these results correctly accord with the literature about exciplex emission of TAPC/TPBi (*J. Phys. Chem. C* **125**, 22809-22816 (2021)).

Furthermore, the emission comes above 550 nm, which is lower energy than the perovskite band gap, which could be attributed to the emission of TAPC electromer (*Appl. Phys. Lett.* **76**,

2352 (2000)) and TPD electromer (*Journal of Luminescence* **130**, 1174-1178 (2010)). Strictly, electromer emission does not include exciplex emission, but we only mentioned exciplex emission in the previous manuscript. Thus, we revised the statement more clearly in the revised manuscript and supplemented the explanation and reference in the explanation for Supplementary Fig. 1,2.

Revision: Page 5 line 1 in the revised manuscript, Supplementary Figs. 1,2, reference and their explanation

Page 5 line 1 in the revised manuscript:

Before: Hole transporting molecules were excluded to avoid exciplex formation in the EL devices (Supplementary Figs. 1-3).

After: Hole transporting molecules were excluded to avoid exciplex formation with the electron transporting layers and parasitic emission (electromer) from the hole-transporting materials in the EL devices (Supplementary Figs. 1-3).

Supplementary Figs. 1,2, reference and their explanation:

Figure S1 | EL spectrum of PeLEDs with hole transporting-CMMs. a, bulk MAPbBr₃ with TAPC. b, bulk MAPbBr₃ with TPD. c, FAPbBr₃ PeNCs with TPD

Figure S2 | PL spectrum of hole transporting-CMM-TPBi mixture films. a, TAPC, TPBi, and mixture of TAPC and TPBi. b, TPD, TPBi, and mixture of TPD and TPBi.

CMMs with electron transporting moieties were studied in the main text. We also tested CMMs with hole transporting materials such as TAPC (di-[4-(N,N-ditolylamino)-phenyl]cyclohexane) and TPD (N,N'-Bis(3-methylphenyl)-N,N'-diphenylbenzidine) using the same fabrication method shown in the main text. All devices incorporating TAPC and TPD showed EL emission from non-perovskite materials (Supplementary Fig. 1). We measured the photoluminescence spectra of TAPC and TPD mixed with TPBi and found that they form exciplex, leading to the broad emission around 470 nm. Since the EML is in contact with TPBi ETL in the device, mixing hole transporting CMMs always leads to the interfacial exciplex formation with TPBi¹. On the other hand, since insulating PFI in the GraHIL prevents the direct contact between PEDOT:PSS hole injection layer and CMMs incorporated in the EML, electron transporting materials can be used as CMM without the formation of interfacial exciplex. **The EL emission with lower energy than that of the perovskite can be attributed to the electromer emission of TAPC and TPD, respectively^{2,3}.**

Reference in the Supplementary Information

1. Park, Y. S. *et al.* Exciplex-forming Co-host for organic light-emitting diodes with ultimate efficiency. *Adv. Funct. Mater.* **23**, 4914–4920 (2013).
2. Nayak, P. K., Patankar, M. P., Narasimhan, K. L. & Periasamy, N. Excited state complex and electroluminescence in TPD-based single layer device. *J. Lumin.* **130**, 1174–1178 (2010).
3. Kalinowski, J., Giro, G., Cocchi, M., Fattori, V. & Di Marco, P. Unusual disparity in electroluminescence and photoluminescence spectra of vacuum-evaporated films of 1,1-bis((di-4-tolylamino) phenyl) cyclohexane. *Appl. Phys. Lett.* **76**, 2352–2354 (2000).

2) It can be seen that the highest EQE are shown at 3.4 V, which the current density is lower than 0.5 mA cm⁻¹. Due to the current density value is too little, EQE measured by integrated sphere method is required to make sure the data are right. By the way, the angular emission profile of the device should be offered to prove the LED is a Lambertian emitter if the authors use a spectroradiometer to measure the EQE. Otherwise, the EQE data should be corrected according the angular emission profile.

Reply: Thank you very much for the detailed comments. At the maximum EQE (for FA_{0.9}GA_{0.1}PbBr₃-TPBi CMM device) where the driving voltage is 3.4V, the luminance is 373 cd/m². This is sufficiently high to be accurately measured with our spectroradiometer setup. We use Konica Minolta's CS2000 spectroradiometer which shows a minimum detectable luminance of 0.075 cd/m² at 0.2° aperture which is much lower than the luminance of our devices at maximum EQE. In addition, our device shows uniform brightness over the whole active area of 4.3 mm² in the whole driving voltage range (Fig. 5e). Therefore, we are completely confident that our device efficiency measurement is correct. In addition, the current density at the maximum EQE is 0.323 mA/cm², and this is also much higher than the detection limit of our Keithley source meter. Moreover, OLED research also reported maximum EQE at low current densities. For example, a green phosphorescent OLED and a green fluorescent OLED published in the literature (Green phosphorescence: *Adv. Funct. Mater.* **23**, 4914-4920 (2018) / Green fluorescence: *Nature* **492**, 234-238 (2012)) reported the maximum EQE at the

current density around 0.1 mA /cm². Therefore, the current density of 0.323 mA/cm² is a sufficiently high level for reliable measurement of device efficiency.

We also obtained the EQE by considering the experimentally measured angle-dependent EL profile (Supplementary Fig. 28). Our device showed a Lambertian emission profile as shown in Supplementary Fig. 28.

- Green phosphorescence OLED (*Adv. Funct. Mater.* **23**, 4914-4920 (2018)): 0.01 mA/cm²

- green fluorescent OLED (*Nature* **492**, 234-238 (2012)): 0.1 mA/cm²

Revision: Page 10 line 13 in the revised manuscript, Reference, Supplementary Fig. 28, and Methods

Page 10 line 13 in the revised manuscript:

Before: The CMM treatment on GA-doped PeNCs results in more efficient PeNC-LED with an EQE of 26.1%, which is the highest efficiency among published reports of PeLEDs using colloidal PeNCs without outcoupling enhancement technique (Fig. 4j and Supplementary Table. 2).

After: The CMM treatment on GA-doped PeNCs results in a more efficient colloidal PeNC-LED with an EQE of 26.1%, **calculated based on angular electroluminescence distribution⁴⁵ (Supplementary Fig. 28). Our colloidal PeNC-LEDs based on this materials design approach achieved** the highest efficiency among published reports of PeLEDs using colloidal PeNCs without an outcoupling enhancement technique (Fig. 5j, Supplementary Fig. 29 and Supplementary Table. 2,3).

Reference

45. Jeong, S. H. *et al.* Characterizing the Efficiency of Perovskite Solar Cells and Light-Emitting Diodes. *Joule* **4**, 1206–1235 (2020).

Supplementary Fig. 28:

Figure S28 | Angle-dependent EL profile for colloidal $\text{FA}_{0.9}\text{GA}_{0.1}\text{PbBr}_3$ PeNC-LED with TPBi.

Method:

Electroluminescence characterization

The current density-voltage-luminescence (J - V - L) characteristics of PeLEDs were measured using a spectroradiometer (CS-2000, Minolta) and an electrical source-measurement unit (Keithley 236). **The EQE of PeLEDs was calculated based on the angle-dependent EL profile⁴⁵.** Transient EL analysis was performed using a streak camera system composed of a streak scope (C10627, Hamamatsu Photonics), a CCD camera (C9300, Hamamatsu Photonics), a delay generator (DG645, Stanford Research Systems), and a function generator (33220A, Agilent).

3) What are the thicknesses of GraHIL and GA-doped perovskite nanocrystal EML? The cross-section transmission electron microscopy (TEM) image of best LED device is required to verify that the parameters of the optical simulation are correct.

Reply: Thank you very much for the beneficial comments. We specified the thickness of each layer in LED devices, as shown in Supplementary Fig. 3. The thickness of GraHIL was accurately measured by the surface profiler. We measured the thickness of PeNC films by ellipsometry measurement (Supplementary Fig. 4). The thickness of GraHIL and GA-doped PeNC layer is 60 nm and 30 nm, respectively. We also added the cross-section SEM image of the GA-doped FAPbBr_3 PeNC-LED instead of the cross-section TEM image, as shown in Supplementary Fig. 27. We tried to measure the cross-section TEM image of the device, but we could not obtain a good image because of the complex condition of focused ion beam and softness of our PeNC films. However, the result of the cross-SEM image is sufficient to identify the thickness of each layer. We verified that the thickness of the PeNC layer is 30 nm, which is consistent with the thickness calculated by ellipsometry measurement. Furthermore, the thicknesses of clearly distinct layers were identical to those shown in Supplementary Fig. 3,

indicating that the thickness used in the optical simulation was correct.

Revision: Supplementary Fig. 27 and its explanation

Figure S27 | Cross-sectional SEM image of colloidal FA_{0.9}GA_{0.1}PbBr₃ PeNC-LED.

To ensure that the thickness of each layer in the device was correct, we measured a cross-SEM image of colloidal FA_{0.9}GA_{0.1}PbBr₃ PeNC-LED. We confirmed that the thickness of FA_{0.9}GA_{0.1}PbBr₃ PeNC is 30 nm, which is consistent with the thickness measured by ellipsometry (Supplementary Fig. 4). In addition, the thickness of the other layers was found to be consistent with the thickness we described (Supplementary Fig. 3).

4) The authors list a table of state-art-of green PeNC LEDs performance in Table S1. In this table, the device in this work is the best performance. In fact, EQEs of green PeNC LEDs over 25% have been reported by others (*Nature* 2022 , 611 , 688-694; *Adv. Mater.* 2023, 2302283; *ACS Energy Lett.* 2023,8, 927-934.). The authors should also list these data in the Table S1, otherwise, they risk misleading the readers that their work is the best result so far.

Reply: Thank you very much for the valuable comment. The reports that the reviewer mentioned are PeLEDs based on bulk perovskite EMLs or using an in-device outcoupling enhancement technique. Bulk perovskite materials are directly crystallized onto the substrate by spin coating the perovskite precursor solution, so this is a completely different approach compared to the colloidal PeNCs (Fig. 1 and Supplementary Fig. 5). Colloidal PeNCs, which are similar to conventional colloidal inorganic nanocrystals such as quantum dots, are synthesized with ligands in a non-polar solvent and their films are deposited using PeNC-dispersed solution, which is already crystallized. Both bulk perovskite and colloidal PeNC system possess their advantages and disadvantages originating from fundamentally different fabrication approaches. Therefore, the two systems are categorized separately when comparing the efficiency of PeLEDs.

The reviewer suggested three examples (*Nature* 2022, 611, 688-694; *Adv. Mater.* 2023,

2302283; ACS Energy Lett. 2023,8, 927-934.). In terms of material approach, *Nature* and *Advanced Materials* papers correspond to the category of bulk perovskite. Therefore, we did not include both literatures previously. Moreover, an *ACS Energy Letter* paper using PeNCs employed an in-device outcoupling enhancement technique with achieving EQE of 26.7%. This technique is not a method of improving the efficiency of the active layer in the conventional device structure. We tried to provide a fair comparison of the device without incorporating any light exaction techniques (Fig. 5j and Supplementary Table. 2). The EQE is defined by the equation below.

$$\text{EQE} = \text{PLQY} \times \text{outcoupling efficiency} \times \text{charge balance factor} \times \text{singlet-triplet ratio}$$

Assuming perfect charge balance and singlet-triplet ratio as 1, the EQE depends on the PLQY and outcoupling efficiency. The conventional LED structure with an isotropic emitter usually has outcoupling efficiency up to nearly 30%, so that the maximum EQE is converged to 30%. However, the outcoupling enhancement technique is the method of increasing the EQE by modulating the optical characteristics of the device structure. For example, a *Nature Photonics* paper reported PeNC-LEDs with an EQE of 23.4%, but EQE was improved up to 45.5% by using an outcoupling hemispherical lens (*Nature Photonics* **15**, 148-155 (2021)). In addition, an *Advanced Materials* paper increased outcoupling efficiency by controlling the charge transporting layer and functional layer, contributing to the achievement of an EQE of 30.84%. Similarly, an *ACS Energy Letter* demonstrated PeNC-LED with an EQE of 26.7% by increasing outcoupling efficiency. However, those outcoupling enhancement techniques are not the approach to developing the inherent luminescence efficiency of perovskite emitters. Therefore, we excluded an *ACS Energy Letter* paper from the table previously. However, we added a new figure comparing EQE including the category of the bulk perovskite and colloidal PeNCs and in-device outcoupling enhancement techniques (Supplementary Fig. 29 and Supplementary Table. 3). Without any outcoupling enhancement techniques, we achieved an EQE of 26.1% through the emitting material design approach alone. Moreover, we achieved the highest efficiency among PeNC-LEDs with a CIEy value of more than 0.75, which is the crucial requirement for next-generation vivid displays.

Revision: Supplementary Fig. 29 and Supplementary Table. 3

Supplementary Fig. 29:

Figure S29 | Summary of reported high-efficiency green-emitting perovskite LEDs based on colloidal PeNC and non-colloidal perovskite (bulk perovskite).

Supplementary Table. 3:

Table S3 | Summary of high-efficiency green-emitting perovskite LEDs based on colloidal PeNC and non-colloidal perovskite (bulk perovskite).

Publication date	Type	Outcoupling enhancement	EL peak (nm)	CIE 1931 color coordinate	Max. EQE (%)	Ref.
2020/07	Colloidal PeNC	No	505	(0.032,0.633) ^a	22	19
2021/01	Colloidal PeNC	No	531	(0.196, 0.764)	23.4	21
2021/05	Bulk perovskite	No	508	(0.049, 0.64)	22.49	24
2021/06	Colloidal PeNC	No	532	(0.19, 0.77)	19.2	22
2021/10	Bulk perovskite	No	514	(0.10, 0.74)	28.1	25
2022/11	Bulk perovskite	No	540	(0.26, 0.72)	28.9	26
2023/01	Colloidal PeNC	Yes	514	(0.05, 0.66)	26.7	27
2023/05	Bulk perovskite	Yes	524	(0.181, 0.743)	30.84	28
-	Colloidal PeNC	No	531	(0.199, 0.762)	26.1	This work

^aEstimated value obtained from digitized data points in reported EL spectrum images

5) How is the stability of PeNC LED with high EQE (26.1%) ? The authors do not demonstrate the result. Is the stability not good?

Reply: We appreciate the reviewer for suggesting this important point. We measured the operational lifetime of colloidal PeNC-LED with high EQE (26.1%) (FA_{0.9}GA_{0.1}PbBr₃ PeNC-TPBi) and compared pristine one (FA_{0.9}GA_{0.1}PbBr₃ PeNC) (Supplementary Fig. 31). As can be seen, the colloidal PeNC-LED with no CMM has the operation lifetime with LT₅₀ of 1.6h. TPBi CMM incorporation led to an improvement of device lifetime up to LT₅₀ of 5h, which is a 3.1-fold enhancement compared to no CMM. Improved operational lifetime by TPBi CMM indicated that lattice-strengthening not only improves luminescent efficiency but also

suppresses the degradation of perovskite.

Revision: Page 10 line 26 in the revised manuscript, Supplementary Fig. 31, and its explanation

Page 10 line 26 in the revised manuscript:

In addition to the improvement of efficiency, the TPBi CMM device showed a longer device half-lifetime by a factor of 3.1 compared to the device without CMM, indicating that lattice-strengthening by TPBi that does not induce ligand detachment also contributes to suppressing the degradation process of the perovskite (Supplementary Fig. 31).

Supplementary Fig. 31 and its explanation:

Figure S31 | Operational lifetime of colloidal FA_{0.9}GA_{0.1}PbBr₃ PeNC devices with CMMs.

To investigate the effect of CMM on the stability of colloidal PeNC-LEDs, we measured the device half-lifetime (LT_{50}) of colloidal FA_{0.9}GA_{0.1}PbBr₃ PeNC-LEDs under the condition that the initial luminance was 100 cd/m². Compared to no CMM device (LT_{50} of 1.6h), the TPBi CMM device showed improved device lifetime (LT_{50} of 5h). On the other hand, PO-T2T exhibited reduced device stability (LT_{50} of 0.7h) because it has highly reactive characteristics with oleic acid ligand (Fig. 6). Therefore, we demonstrated that lattice-strengthening by TPBi CMM that does not induce ligand detachment not only enhances luminescent efficiency, but suppresses the degradation of perovskite.

6) The TEM images of perovskite nanocrystals are suggested to offered by the autho

Reply: Previously, we inquired to the editor about this question because of the typo in the term “autho”. We thought that “autho” in this sentence was a typo for “authors”. However, we received a modified version of the sentence below from the editor.

“The cross-sectional TEM image of LED device is suggested to offered by the authors.”

Since this question is a complete duplicate of the #3 comment suggested by the reviewer, we are still not sure whether this question was asking for “TEM images of PeNC materials” or “a cross-TEM image of the PeNC-LED”. Nonetheless, in order to prevent further bothering, we

would like to respond to both questions and greatly appreciate your understanding of this situation.

If this question is intended to measure the cross-TEM image of colloidal PeNC-LED, we would really appreciate it if the reviewer referred to the #3 comment-response.

In case this question was intended to measure TEM images of colloidal PeNCs, we additionally conducted TEM analysis for colloidal FAPbBr₃ PeNCs and FA_{0.9}GA_{0.1}PbBr₃ PeNCs with CMMs (Fig. 4b, Supplementary Figs. 12, 21).

Actually, the colloidal PeNCs are substantially beam-sensitive to the electron beam in TEM, showing significant changes in morphologies at high accumulated doses (*Science* **359**, 675-679 (2018)). Besides, organic cation-based colloidal PeNC materials showed more vulnerable characteristics to beam damage (*Chem* **8**, 327-339 (2022)). Therefore, in order to obtain good images as much as possible, we utilized a TEM instrument equipped with a cold field emission gun, an image corrector, and a Gatan K3 IS direct electron detector. Additionally, we also delicately found the regions of interest and adjusted image focus as quickly as possible with a small electron dose rate of $\sim 1 \text{ e}^-/\text{Å}^2\cdot\text{s}$, and measured TEM images with an electron dose of $10 \text{ e}^-/\text{Å}^2\cdot\text{s}$ using a K3 IS detector at optimized condition with negligible structural changes of nanocrystals while maintaining the signal-to-noise ratio of the images.

As can be seen in Fig. 4b and Supplementary Figs. 12,21, colloidal FAPbBr₃ PeNCs with no CMM, 3TPYMB, PO-T2T, and TPBi showed average particle sizes of 9.78 nm, 9.90 nm, 9.98 nm, and 9.88 nm, respectively. Colloidal FA_{0.9}GA_{0.1}PbBr₃ PeNCs with no CMM, PO-T2T, and TPBi showed average particle sizes of 9.94 nm, 9.75 nm, and 9.94 nm, respectively. These results demonstrate that CMMs do not significantly affect the average particle size and particle size distribution.

Revision: Page 8 line 19 in the revised manuscript, Fig. 4b, Supplementary Fig. 12, Page 9 line 29 in the revised manuscript, Supplementary Fig. 21

Page 8 line 19 in the revised manuscript, Fig. 4b, and Supplementary Fig. 12:

High-resolution transmission electron microscopy (TEM) images of colloidal FAPbBr₃ PeNCs showed average particle size of ~ 9.78 nm without CMM, 9.90 nm with 3TPYMB, 9.98 nm with PO-T2T, and 9.88 nm with TPBi (Fig. 4b and Supplementary Fig. 12). This result indicates that post-synthesis blending of CMMs into colloidal PeNCs has negligible effect on the average particle size and distribution.

Fig. 4 | Interaction between FAPbBr₃ PeNC and CMMs. a-c, Independent characteristics of FAPbBr₃ PeNCs with CMMs. The size distribution (**b**) of colloidal FAPbBr₃ PeNCs. Particle size was extracted from transmission electron microscope (TEM) images.

Figure S12 | TEM images of colloidal FAPbBr₃ PeNCs with CMMs. a, no CMM, **b,** 3TPYMB, **c,** PO-T2T and **d,** TPBi. Each image size is 40 nm × 40 nm.

Page 9 line 29 in the revised manuscript and Supplementary Fig. 21:

Similar to the studies of FAPbBr_3 PeNCs (Fig. 4a,b and Supplementary Figs. 12,13), CMM incorporation had negligible impact on the crystal structure, film morphology, and particle size of GA-doped FAPbBr_3 PeNCs (Supplementary Figs.19-21).

Figure S21 | TEM images of colloidal $\text{FA}_{0.9}\text{GA}_{0.1}\text{PbBr}_3$ PeNCs with CMMs. a, no CMM, b, PO-T2T, c, TPBi, and d, size distributions. Each image size is 40 nm \times 40 nm.

REVIEWER COMMENTS

Reviewer #1 (Remarks to the Author):

The authors have made significant efforts addressing my comments about the material characterizations.

However, the LED lifetime study is not convincing. Although the authors observed a longer T50 with TPBi, the mechanism is still unclear. The degradation curves of all three devices are identical, suggesting the EL all degrade at the same rate, likely from the same causes. I do not understand why this indicates a slower ligand detachment. The data graph look doubtful. The data density of the first hour of the TPBi curve has way less points than that of the control device, e.g., only 3 points in the first 20min (0.1 hr). This has to be carefully looked into!

Reviewer #2 (Remarks to the Author):

I am glad to see that the authors addressed the various comments from all reviewers. The additional experimental work now sustain the various claims of the authors, although I am still not entirely convinced I would call the CMM effect as "Lattice-strengthening" considering the additive only interact with the A cations. This is a rather semantic problem, thus trivial and it does not prevent publication of this manuscript in my view.

The manuscript presents a novel approach to suppress lattice disorder, as such it contains findings which are of interest to the nanomaterials community. In addition, the efficacy of such treatment is demonstrated by the performance of the respective LEDs. Therefore, I suggest the manuscript for acceptance.

I have some minor comments, which you can find here below:

Page 8, "High-resolution transmission electron microscopy (TEM) images of colloidal FAPbBr₃ PeNCs showed average particle size of ~9.78 nm without CMM, 9.90 nm with 3TPYMB, 9.98 nm with PO-T2T, and 9.88 nm with TPBi (Fig. 4b and Supplementary Fig. 12)." Please estimate the error on these values. Also, in other part of the text errors/error bars are not reported (see for example "grain size" in Fig. S14)-

Stability data in figure S31. Can the author compare the stability of their devices with literature? Is this operational lifetime comparable to state-of-the-art perovskite NC LEDs? It would be great to include a table for comparison and maybe a very brief discussion.

Figure S16, since the y-axis is in arbitrary units, it cannot be "Absorbance" (this is a physical property measured in optical density) please change the y-axis label to "Optical Absorption".

Figure S25, see previous comments on lack of error (bars, for PLQY) and "Absorbance" label.

Reviewer #3 (Remarks to the Author):

The authors have carried out the relevant experiments and answered the relevant questions

one by one. The corresponding scientific questions have been cleared. I think the article could be published in nature communications now.

Response to the Reviewers' Comments

[Nature Communications NCOMMS-23-31521-B]: Surface-Binding Molecular Multipods Strengthen the Halide Perovskite Lattice and Boost Luminescence

We would like to thank the reviewers and the editors for their valuable comments on our work. Such productive feedbacks from the world experts in this field helped us improve our manuscript. We provide a point by point response below to the reviewers' questions. We also revised our manuscript to comply with the reviewers' comments and have highlighted the revised parts in red.

Reviewer #1 (Remarks to the Author):

The authors have made significant efforts addressing my comments about the material characterizations. However, the LED lifetime study is not convincing. Although the authors observed a longer T50 with TPBi, the mechanism is still unclear. The degradation curves of all three devices are identical, suggesting the EL all degrade at the same rate, likely from the same causes. I do not understand why this indicates a slower ligand detachment. The data graph look doubtful. The data density of the first hour of the TPBi curve has way less points than that of the control device, e.g., only 3 points in the first 20min (0.1 hr). This has to be carefully looked into!

Reply: Thank you very much for providing detailed comments. In the previous operational lifetime curve of PeNC-LEDs, we plotted the time scale (x-axis) in a log scale. This way is generally used when the difference in operational lifetime (LT_{50}) is very large by orders of magnitudes (Nature Nanotechnology (2024), doi: 10.1038/s41565-023-01581-2). We speculate that because the time scale of the operational lifetime curve was plotted in a log scale, it appeared that there were fewer data points in the initial region (<0.1 h). To clear up the misunderstanding, we provide an operational lifetime curve magnifying the initial region of the time (<0.5 h) (Fig. R1). As can be seen in Figure R1, the number of data points is sufficient to identify the degradation characteristics of them.

Figure R1. Operational lifetime curve of colloidal FA_{0.9}GA_{0.1}PbBr₃ PeNC LEDs in the region below 0.5 h.

However, there were differences in the time intervals between conditions. The data interval of no CMM device was 0.005h while that of PO-T2T and TPBi devices was 0.015h. The electrical stress equivalent to 100 cd/m² is applied consistently, and the only difference is the interval at which data is collected. Simply varying the collecting data interval does not affect the degradation characteristics in operational stability analysis except in cases where the interval is set extremely wide. However, we strongly believe that the data intervals of 0.005h and 0.015h are sufficient to analyze the degradation characteristics of our devices (Supplementary Fig. 31 and Fig. R1).

As the same with the references published by our research group [*Nature* **611**, 688-694 (2022) / *Nature Nanotechnology* **17**, 590-597 (2022), *Nature Photonics* **15**, 148-155 (2021)], operational lifetime of LED devices was measured by using constant current mode with setting initial luminance (measurement condition in this manuscript is L₀= 100 cd/m²). We also compared the operational lifetime of our no CMM device with recently reported literature published by our research group [*Nature Nanotechnology* (2024), doi: 10.1038/s41565-023-01581-2] as shown below (Fig. R2). Both devices showed similar device lifetime curves, further supporting that our data in this manuscript were reliable.

Figure R2. Operational lifetime of PeNC-LEDs of no CMM device and reference.

To prevent misunderstanding or doubt from the potential readers, we revise this curve by plotting the x-axis as a linear scale (Supplementary Fig. 31). As can be seen below, the three conditions have clearly different degradation characteristics. Compared to no CMM device (LT₅₀ of 1.6 h), the TPBi CMM device showed significantly improved operational lifetime (LT₅₀ of 5h). PO-T2T CMM device, on the other hand, showed a reduced operational lifetime (LT₅₀ of 0.7h). Our study explained that although PO-T2T induces lattice-strengthening (Fig. 3a-f and Fig. 5b-c), the ligand detachment occurs severely in FA_{0.9}GA_{0.1}PbBr₃ PeNC-PO-T2T system due to highly nucleophilic group of phosphine oxide group in PO-T2T, canceling out lattice-strengthening effect (Fig. 6). In fact, we proved that the luminescent efficiency of FA_{0.9}GA_{0.1}PbBr₃ PeNC LED with PO-T2T CMM was reduced compared to no CMM device (Fig. 5d-j).

Figure S31 | Operational lifetime of colloidal FA_{0.9}GA_{0.1}PbBr₃ PeNC devices with CMMs.

We also tried to compare the EL degradation rate using the stretched exponential decay (SED) function [*Nature Photonics* **15**, 630-634 (2021)]. The most basic function is single exponential decay [$L/L_0 = \exp(-t/\tau)$]. If various factors contribute to degradation in EL devices, the EL decay curve deviates from the usual exponential, and in this case, the SED function can be used as an alternative [*Nature Photonics* **15**, 630-634 (2021)].

$$L/L_0 = \exp(-t/\tau)^\beta$$

(L_0 = initial luminance. t = time, τ = decay time constant, and β = dispersion factor)

We fitted our operational lifetime curve using SED function and obtained physical property parameters as shown below (Figure R3 and Table R1). Since this field of perovskite LEDs has been studied for a short period of time compared to organic LEDs (OLEDs), we cannot yet analyze the decay curve of our devices with complete accuracy, but these SED fittings allow us to compare decay time constant (τ). Obviously, a TPBi CMM device exhibited a higher

decay time constant ($\tau_{TPBi} = 6.89$ h) compared to no CMM ($\tau_{no\ CMM} = 2.37$ h). Furthermore, a PO-T2T CMM device showed a reduced decay time constant of 0.93h. Those results clearly indicate that a PO-T2T CMM device has a faster degradation rate than no CMM device.

Figure R3 | Operational lifetime curve of FA_{0.9}GA_{0.1}PbBr₃ PeNC-LEDs with stretched exponential fitting applied. $L/L_0 = \exp(t/\tau)^\beta$.

Condition	τ (h)	β
no CMM	2.37	1.01
PO-T2T	0.93	0.88
TPBi	6.89	1.08

Table R1 | Fitted parameters of operational lifetime curve with stretched exponential fitting applied. $L/L_0 = \exp(t/\tau)^\beta$.

To further solidify our claim, we additionally measured the photostability of FA_{0.9}GA_{0.1}PbBr₃ PeNC films embedding with CMMs (Supplementary Fig. 32). We fabricated FA_{0.9}GA_{0.1}PbBr₃ PeNC films on the quartz substrate and encapsulated them. We applied constant illumination to FA_{0.9}GA_{0.1}PbBr₃ PeNC films using a 405 nm continuous-wave laser and measured the PL spectra by using a CCD spectrometer. As shown in Supplementary Fig. 32, the film without CMMs decreased to 50% of its initial PL intensity after 48 min. TPBi CMM showed much-improved photostability, remaining 50% of its initial PL intensity after 125 min. Such improvement clearly indicates that lattice-strengthening by TPBi CMM enhances the stability of colloidal FA_{0.9}GA_{0.1}PbBr₃ PeNCs. On the contrary, PO-T2T CMM showed lower photostability than no CMM film. This reduction of stability could be attributed to a strong ligand detachment effect due to the phosphine oxide group (Fig. 6).

We already demonstrated that PO-T2T has a strong interaction with oleic acid, leading to reduced luminescent efficiency of FA_{0.9}GA_{0.1}PbBr₃ PeNC films and LED devices (Fig. 5). We also previously confirmed that PO-T2T induced metallic Pb formation and reduced exciton binding energy in FA_{0.9}GA_{0.1}PbBr₃ PeNCs (Fig. 6a-c). We verified that the detrimental ligand

detachment effect by a PO-T2T CMM prevails against the lattice-strengthening effect because of the strong interaction between the phosphine oxide group and oleic acid.

Likewise, photostability of film with PO-T2T CMM was dramatically reduced, which could be attributed to the ligand detachment effect (Supplementary Fig. 31). In addition, the photostability characteristics of the $\text{FA}_{0.9}\text{GA}_{0.1}\text{PbBr}_3$ PeNC films with CMMs showed nearly the same trends as the operational stability of the devices, further supporting that the reduced operational lifetime of PO-T2T CMM device is related to ligand detachment effect.

Figure S32 | Photostability of $\text{FA}_{0.9}\text{GA}_{0.1}\text{PbBr}_3$ PeNC films with CMMs.

Reviewer #2 (Remarks to the Author):

I am glad to see that the authors addressed the various comments from all reviewers. The additional experimental work now sustain the various claims of the authors, although I am still not entirely convinced I would call the CMM effect as "Lattice-strengthening" considering the additive only interact with the A cations. This is a rather semantic problem, thus trivial and it does not prevent publication of this manuscript in my view.

The manuscript presents a novel approach to suppress lattice disorder, as such it contains findings which are of interest to the nanomaterials community. In addition, the efficacy of such treatment is demonstrated by the performance of the respective LEDs. Therefore, I suggest the manuscript for acceptance.

I have some minor comments, which you can find here below:

1) Page 8, "High-resolution transmission electron microscopy (TEM) images of colloidal FAPbBr₃ PeNCs showed average particle size of ~9.78 nm without CMM, 9.90 nm with 3TPYMB, 9.98 nm with PO-T2T, and 9.88 nm with TPBi (Fig. 4b and Supplementary Fig. 12)." Please estimate the error on these values. Also, in other part of the text errors/error bars are not reported (see for example "grain size" in Fig. S14)-

Reply: Thank you for the valuable comments. As the reviewer commented, we calculated the standard error values of the average sizes and marked them on the corresponding figures. Colloidal FAPbBr₃ PeNCs with no CMM, 3TPYMB, PO-T2T, and TPBi showed the average particle size of 9.78 ± 0.23 nm, 9.90 ± 0.21 nm, 9.98 ± 0.20 nm, and 9.88 ± 0.24 nm, respectively (Fig. 4b and Supplementary Fig. 21). Also, colloidal FA_{0.9}GA_{0.1}PbBr₃ PeNCs with no CMM, PO-T2T, and TPBi showed the average particle size of 9.94 ± 0.22 nm, 9.75 ± 0.23 nm, and 9.94 ± 0.22 nm, respectively (Supplementary Fig. 21). All colloidal PeNCs without and with CMMs showed standard error values lower than 0.3 nm, indicating that narrow size distribution was maintained even with the addition of CMMs. Bulk MAPbBr₃ with no CMM, 3TPYMB, PO-T2T and TPBi showed the average grain size of 159.34 ± 5.29 nm, 86.13 ± 1.74 nm, 96.24 ± 1.43 nm, and 84.04 ± 1.54 nm, respectively. The reduction of the average grain size and standard error by incorporation of CMMs in bulk MAPbBr₃ is due to the nanocrystal pinning effect [*Science* **350**, 1222-1225 (2015)].

Revision: Page 8 line 21 in the revised manuscript, Fig. 4b, Supplementary Fig. 14, Supplementary Fig. 21,

Page 8 line 21 in the revised manuscript and Fig. 4b:

High-resolution transmission electron microscopy (TEM) images of colloidal FAPbBr₃ PeNCs showed

average particle size of $\sim 9.78 \pm 0.23$ nm without CMM, 9.90 ± 0.21 nm with 3TPYMB, 9.98 ± 0.20 nm with PO-T2T, and 9.88 ± 0.24 nm with TPBi (Fig. 4b and Supplementary Fig. 12).

b

Fig. 4 | Interaction between FAPbBr₃ PeNC and CMMs. a-c, Independent characteristics of FAPbBr₃ PeNCs with CMMs. The size distribution (**b**) of colloidal FAPbBr₃ PeNCs. Particle size was extracted from transmission electron microscope (TEM) images. **Average particle size and standard error were calculated.**

Supplementary Fig. 14:

Figure S14 | SEM images of bulk MAPbBr₃ films embedding CMMs with grain size distribution. a, no CMM, b, 3TPYMB, c, PO-T2T and d, TPBi. Average grain size and standard error were calculated.

To investigate the morphological effect of CMMs on bulk MAPbBr₃, we measured SEM images of CMM-embedded bulk MAPbBr₃ films. Bulk MAPbBr₃ with no CMM showed an average grain size of 159.34 ± 5.29 nm. Because CMMs are embedded into perovskite during the crystallization process (additive-based nanocrystal pinning effect)⁴, the incorporation of 3TPYMB, PO-T2T, and TPBi reduced average grain size to 86.13 ± 1.74 nm, 96.24 ± 1.43 nm, and 84.04 ± 1.54 nm, respectively, which are very similar. We could not find a clear correlation between morphology and luminescent efficiency of bulk MAPbBr₃ upon incorporation of various CMM (Fig. 2 and Supplementary Fig. 8d-f).

Supplementary Fig. 21:

Figure S21 | TEM images of colloidal $\text{FA}_{0.9}\text{GA}_{0.1}\text{PbBr}_3$ PeNCs with CMMs. a, no CMM, b, PO-T2T, c, TPBi, and d, size distributions. Each image size is $40 \text{ nm} \times 40 \text{ nm}$. Average particle size and standard error were calculated.

2) Stability data in figure S31. Can the author compare the stability of their devices with literature? Is this operational lifetime comparable to state-of-the-art perovskite NC LEDs? It would be great to include a table for comparison and maybe a very brief discussion.

Reply: Thank you for the constructive comments. In the previous manuscript, we compared EQE versus CIEy for state-of-the-art green-emitting colloidal PeNC LEDs (Fig. 5j) and summarized the CIE coordinate, EL peak, FWHM, and EQE in Table S2. Here, we newly

added a column showing the operational lifetime of state-of-the-art PeNC-LEDs including the condition of initial luminance (L_0). Furthermore, we also added two papers reporting high EQE published during the revision period of this manuscript (Ref. 24, 25) while updating the data point in Fig. 5j. As can be seen in Table S2, the operational lifetime of our PeNC-LEDs with TPBi CMMs ($LT_{50} \sim 5$ h) is comparable to that of the literature. We also added a brief discussion related to the operational stability of PeNC-LEDs including ours in the section of Table S2.

Revision: Table. S2 and Fig. 5j

Before:

Table S2:

Table S2 | Summary of evolved performance of state-art-of green-emitting colloidal PeNC LEDs without outcoupling enhancement technique and their EL emission characteristics (CIE 1931 color coordinate, EL peak, FWHM, Max. EQE).

Publication date	CIE 1931 color coordinate	EL peak (nm)	FWHM (nm)	Max. EQE (%)	Ref.
2015/07	(0.061, 0.785) ^a	516	23	0.12	8
2015/12	(0.14, 0.77)	524	24	1.1	9
2016/05	(0.066, 0.764) ^a	525	24	1.06	10
2016/08	(0.085, 0.770) ^a	515	19	3.0	11
2016/11	(0.05, 0.71)	512	20	6.27	12
2017/08	(0.168, 0.773)	529	22.8	1.43	13
2018/03	(0.20, 0.76)	532	25	13.4	14
2018/09	(0.0613, 0.732)	519	18	6.04	15
2018/10	(0.174, 0.78)	532	22	3.53	16
2018/10	(0.09, 0.71)	518	18	16.48	17
2019/06	(0.172, 0.765)	528	24	2.96	18
2020/07	(0.032, 0.633) ^a	505	17.3 ^a	22	19
2020/08	(0.074, 0.707)	516	20	18.7	20
2021/01	(0.196, 0.764)	531	20.7	23.4	21
2021/06	(0.19, 0.77)	532	21	19.2	22

2022/05	(0.187,0.767)	530	20	23.3	23
-	(0.199, 0.762)	531	20.7	26.1	This work

^aEstimated value obtained from digitized data points in reported EL spectrum images

Fig. 5j:

Fig. 5 | Effect of CMMs on $\text{FA}_{0.9}\text{GA}_{0.1}\text{PbBr}_3$ PeNCs and their luminescence characteristics. j, Summary of reported EQE according to CIEy of green-emitting colloidal PeNC-LEDs without outcoupling enhancement technique (Inset: corresponding color coordinate of our device in CIE 1931 color space).

After:

Table S2:

Table S2 | Summary of evolved performance of state-of-the-art green-emitting colloidal PeNC LEDs without outcoupling enhancement technique and their EL emission characteristics (CIE 1931 color coordinate, EL peak, FWHM, Max. EQE, and device lifetime).

Publication date	CIE 1931 color coordinate	EL peak (nm)	FWHM (nm)	Max. EQE (%)	Device lifetime (LT_{50})	Ref.
2015/07	(0.061, 0.785) ^a	516	23	0.12	-	8
2015/12	(0.14, 0.77)	524	24	1.1	-	9
2016/05	(0.066, 0.764) ^a	525	24	1.06	10 day ($\text{L}_0 = \text{unknown}$)	10
2016/08	(0.085, 0.770) ^a	515	19	3.0	-	11
2016/11	(0.05, 0.71)	512	20	6.27	-	12

2017/08	(0.168, 0.773)	529	22.8	1.43	-	13
2018/03	(0.20, 0.76)	532	25	13.4	800 s ($L_0 = 105 \text{ cd/m}^2$)	14
2018/09	(0.0613, 0.732)	519	18	6.04	-	15
2018/10	(0.174, 0.78)	532	22	3.53	32 min ($L_0 = 14.2 \text{ cd/m}^2$)	16
2018/10	(0.09, 0.71)	518	18	16.48	136 min ($J = 0.6 \text{ mA/cm}^2$)	17
2019/06	(0.172, 0.765)	528	24	2.96	37 h ^b	18
2020/07	(0.032, 0.633) ^a	505	17.3 ^a	22	60 min ($L_0 = 1200 \text{ cd/m}^2$)	19
2020/08	(0.074, 0.707)	516	20	18.7	15.8 h ^b	20
2021/01	(0.196, 0.764)	531	20.7	23.4	132 min ($L_0 = 100 \text{ cd/m}^2$)	21
2021/06	(0.19, 0.77)	532	21	19.2	20 min ($L_0 = 100 \text{ cd/m}^2$)	22
2022/05	(0.187, 0.767)	530	20	23.3	81 min ($L_0 = 100 \text{ cd/m}^2$)	23
2023/09	(0.097, 0.781) ^a	518	17.8 ^a	23.45	77 s ($L_0 = 220 \text{ cd/m}^2$)	24
2023/10	(0.092, 0.766) ^a	517	16.1	24.13	54 min ($L_0 = 10,000 \text{ cd/m}^2$)	25
-	(0.199, 0.762)	531	20.7	26.1	5h ($L_0 = 100 \text{ cd/m}^2$)	This work

^aEstimated value obtained from digitized data points in reported EL spectrum images

^bExtrapolated to 100 cd/m² using acceleration factor (n), $L_0^n T_{50} = \text{constant}$

We demonstrated the highest EQE of 26.1% among green-emitting colloidal PeNC-LEDs without relying on outcoupling enhancement techniques. Moreover, a lattice-strengthening CMM without ligand detachment effect enabled improvement in the real-time operational lifetime of PeNC-LEDs up to LT₅₀ of 5h, comparable to real-time lifetimes of state-of-the-art PeNC-LEDs. However, the overall operational lifetime of state-of-the-art PeNC-LEDs is far behind compared to that of bulk polycrystalline PeLEDs²⁶. This gap stands out the necessity to further develop properties of colloidal PeNCs by designing additive molecules and strongly surface-binding ligands that can ameliorate the stability of PeNCs. Our comprehensive understanding of the dynamic disorder of perovskite lattice and dynamic binding characteristics of the ligands in colloidal PeNCs would be a stepping stone to developing future strategies for improving the luminescent efficiency and operational lifetime of PeNC-LEDs.

Fig. 5j:

Fig. 5 | Effect of CMMs on $\text{FA}_{0.9}\text{GA}_{0.1}\text{PbBr}_3$ PeNCs and their luminescence characteristics. j, Summary of reported EQE according to CIEy of green-emitting colloidal PeNC-LEDs without outcoupling enhancement technique (Inset: corresponding color coordinate of our device in CIE 1931 color space).

3) Figure S16, since the y-axis is in arbitrary units, it cannot be “Absorbance” (this is a physical property measured in optical density) please change the y-axis label to “Optical Absorption”.

Reply: Thank you very much for the detailed comments. We measured UV-vis absorption spectra of CMM films and perovskite films with CMMs. As the reviewer commented, we changed the y-axis label from Absorbance (a.u.) to Optical Absorption (a.u.).

Revision: Supplementary Fig. 16

Before:

Figure S16 | UV-vis absorption spectra. a, CMM films, b-d, Perovskite films with CMMs. bulk MAPbBr₃ (b), FAPbBr₃ PeNCs (c), FA_{0.9}GA_{0.1}PbBr₃ PeNCs (d).

After:

Figure S16 | UV-vis absorption spectra. a, CMM films, **b-d**, Perovskite films with CMMs. bulk MAPbBr₃ (**b**), FAPbBr₃ PeNCs (**c**), FA_{0.9}GA_{0.1}PbBr₃ PeNCs (**d**).

4) Figure S25, see previous comments on lack of error (bars, for PLQY) and “Absorbance” label.

Reply: We appreciate the reviewer for presenting the comment to solidify our experimental results. In the previous Supplementary Fig. 25a, we used only one data for each condition. Through additional experiments measuring the PLQY of PeNC-TPBi solutions, we calculated average PLQY values and standard error. As a result, we revised Supplementary Fig. 25a showing average PLQY values with error bars (standard error).

In previous Supplementary Fig. 25b, the y-axis label was “Absorbance (a.u.)”. At this time, it was an error to mark it as an arbitrary unit. Here, we used raw data for absorbance of FA_{0.9}GA_{0.1}PbBr₃ PeNC solutions containing TPBi, to exhibit that increasing concentration of TPBi solution resulted in a higher composition of TPBi in PeNC-TPBi mixed solution. As can be seen in Supplementary Fig. 25b, increasing the concentration of TPBi solution resulted in gradually increased absorbance in the region below 350 nm. Therefore, we revised the y-axis label from “Absorbance (a.u.)” to “Absorbance”, because we used raw data as was and absorbance is a dimensionless quantity.

Revision: Supplementary Fig. 25

Before:

Figure S25 | Effect of TPBi CMM on colloidal FA_{0.9}GA_{0.1}PbBr₃ PeNC solution depending on the concentration of TPBi solution. a, PLQY. **b**, UV-vis absorption spectra. **c**, PL spectrum.

After:

Figure S25 | Effect of TPBi CMM on colloidal FA_{0.9}GA_{0.1}PbBr₃ PeNC solution depending on the concentration of TPBi solution. a, PLQY. The error bars are the standard error. b, UV-vis absorption spectra. c, PL spectrum.

REVIEWERS' COMMENTS

Reviewer #1 (Remarks to the Author):

The authors have addressed my concerns, and I have no further comment, and support its publication.

Reviewer #2 (Remarks to the Author):

In my opinion, the manuscript can be published as is.